# Natural Coumarin Shows Toxicity to *Spodoptera litura* by Inhibiting Detoxification Enzymes and Glycometabolism

**DOI:** 10.3390/ijms241713177

**Published:** 2023-08-24

**Authors:** Tao Xia, Yan Liu, Zhanjun Lu, Haizhong Yu

**Affiliations:** 1College of Life Sciences, Gannan Normal University, Ganzhou 341003, China; xiatao19980326@163.com (T.X.); maryann2020@163.com (Y.L.); luzhanjun7@139.com (Z.L.); 2National Navel Orange Engineering Research Center, Gannan Normal University, Ganzhou 341003, China

**Keywords:** *Spodoptera litura*, coumarin, transcriptome sequencing, metabolome sequencing, RNA interference

## Abstract

Coumarin and its derivatives are plant-derived compounds that exhibit potent insecticidal properties. In this study, we found that natural coumarin significantly inhibited the growth and development of *Spodoptera litura* larvae through toxicological assay. By transcriptomic sequencing, 80 and 45 differentially expressed genes (DEGs) related to detoxification were identified from 0 to 24 h and 24 to 48 h in *S. litura* after coumarin treatment, respectively. Enzyme activity analysis showed that CYP450 and acetylcholinesterase (AChE) activities significantly decreased at 48 h after coumarin treatment, while glutathione *S*-transferases (GST) activity increased at 24 h. Silencing of *SlCYP324A16* gene by RNA interference significantly increased *S. litura* larval mortality and decreased individual weight after treatment with coumarin. Additionally, the expression levels of DEGs involved in glycolysis and tricarboxylic acid (TCA) cycle were inhibited at 24 h after coumarin treatment, while their expression levels were upregulated at 48 h. Furthermore, metabonomics analysis identified 391 differential metabolites involved in purine metabolism, amino acid metabolism, and TCA cycle from 0 to 24 h after treated with coumarin and 352 differential metabolites associated with ATP-binding cassette (ABC) transporters and amino acid metabolism. These results provide an in-depth understanding of the toxicological mechanism of coumarin on *S. litura*.

## 1. Introduction

*Spodoptera litura* (Lepidoptera: Noctuidae) is a globally destructive pest that damages multiple host plants and causes enormous economic losses [1]. It has three developmental stages, including the larval, pupal, and adult phases. The larval stage is the most critical for feeding and crop damage [2,3]. Currently, *S. litura* is controlled primarily using chemical insecticides. However, the inappropriate use of insecticides leads to adverse effects on human health and environment sustainability [4,5]. Therefore, developing pollution-free and environmentally friendly biocontrol strategies are urgently needed.

Plants produce a variety of secondary metabolites to protect themselves against predators such as herbivores and pathogens [6]. Coumarin is a phenolic compound that metabolizes into a toxic compound and affects the gut [7]. In plants, phenolic compounds are one of the most common and widespread groups of secondary metabolites [8]. The oxidation of phenol is catalyzed by polyphenol oxidase and peroxidase, a potential defense mechanism in plants against insect herbivores [9]. Recent studies have demonstrated that natural coumarin and its derivatives exhibit strong insecticidal activity [10]. Poudel et al. reported that coumarin had a toxic effect on *Drosophila melanogaster* through activation of the aversive gustatory receptor neurons (GNRs), and that gustatory receptor 33 (GR33) is required for oviposition to avoid coumarin-laced food [11]. Interestingly, GR180, a functional receptor responding to coumarin which is highly expressed in the maxillary galea of *Helicoverpa armigera*, was also involved in sensing sinigrin and strychnine [12]. These results indicated that coumarin might affect the expression of chemosensory receptors, and thus would influence the feeding choice of insects. Additionally, scoparone, a well-known phenolic coumarin, exhibited potent acaricidal activity against *Tetranychus cinnabarinus*. Further research revealed that scoparone can target the interface between calmodulin 1 (CaM1) and N-type voltage-gate Ca^2+^ channel (VGCC) and activate the CaM-binding site [13,14]. However, the toxicological mechanism of coumarin against *S. litura* remains unclear.

Insect midguts play an important role in digestion and nutrient absorption, as well as serve as the target of pathogenic microorganisms, pesticides, and various toxins [15]. Insect glycometabolism does not allow them to directly absorb polysaccharides and oligosaccharides from food; instead, they require decomposition into monosaccharides before they may be utilized [16]. The TCA cycle is an important pathway in terms of energy metabolism responsible for the oxidation of respiratory substrates to produce adenosinetriphosphate (ATP) [17]. Some exogenous substances, including pesticides and alkaloids, significantly inhibit glycometabolism and energy metabolism in insects. For example, in *Bombyx mori*, sanguinarine impaired trehalose hydrolysis and reduced trehalase activity, which resulted in the inhibition of energy metabolism [18]. Wang et al. revealed that the glycolysis and the TCA cycle showed significant responses to fenpropathrin in *B. mori* [19]. Additionally, the midgut also plays a major role in insecticide resistance [20]. Most detoxification-related enzymes are produced and secreted in the midgut of insects, including in *Anoplophora glabripennis* and *Plutella xylostella* [21,22]. In insects, the detoxification process occurs in three phases: phase I, phase II, and phase III, among which phase I and phase II involve metabolic enzymes including cytochrome P450 monooxygenases (CYPs), glutathione *S*-transferases (GSTs), and esterases (ESTs) [23]. Insect CYP is one of the most prominent enzyme superfamilies involved in detoxifying xenobiotics by directing the nucleophilic functional group to the xenobiotic compound [24]. GSTs are important in metabolizing endogenous compounds and xenobiotics. GSTs can catalyze the conjugation of reduced glutathione with xenobiotics, making the conjugated compounds soluble and enabling their detoxification through excretion [25,26,27]. Carboxy/cholinesterases (CCEs) are a major class of detoxification genes in all living organisms involved in many metabolic reactions [28]. In *Sitobion avenae*, CarE and GST activities were positively correlated with the concentration of secondary plant compounds in artificial diets [29]. Shu et al. revealed that camptothecin, a bioactive secondary metabolite from *Camptotheca acuminata*, significantly disrupted the detoxification pathways in *Spodoptera frugiperda* [30]. These results suggested that detoxification enzymes are effective insecticidal targets for plant secondary compounds.

In recent years, omics techniques have provided insights into the complex physiological responses in individuals caused by exogenous substances. Metabolomics can be performed to identify and quantify metabolites, while transcriptomics can provide important information about gene expression [31]. In this study, a toxicological test was performed to determine the effects of coumarin on the growth and development of *S. litura*. A total of 2906 and 1492 DEGs were identified from 0 h to 24 h and 24 h to 48 h after treatment with coumarin by comparative transcriptomics, respectively. Functional analysis revealed that DEGs involved in glycolysis and TCA cycle were downregulated at 24 h after coumarin treatment, while they were upregulated at 48 h. Furthermore, the activities of the enzyme, CYP450, AChE, and GST, were measured. Furthermore, inhibition of the *SlCYP324A16* gene significantly increased the sensitivity to coumarin in *S. litura* larvae.

## 2. Results

### 2.1. Toxicological Analysis of S. litura Response to Coumarin

The toxicity experiment was conducted to assess the effect of coumarin on *S. litura* larval growth and development. The results showed that different concentrations (1%, 2%, and 3%) of coumarin significantly inhibited *S. litura* larval growth and development (Figure 1; Table 1). Before coumarin treatment (day 0), the average *S. litura* larval weight was approximately 12 mg (Table 1). With the extension of days, *S. litura* larval weight increased quickly from day 1. In the control groups, the weight of larvae on the 5th day (day 5) was approximately 26 times that of day 0 (Figure 1A; Table 1). However, the larvae of the coumarin-treated groups appeared feeble, with their growth and development inhibited from day 1 and day 5. Increase in the concentration of coumarin, had an inhibitory effect on *S. litura* larvae. Especially on day 5, the weight of *S. litura* larvae in the control groups was 12 times that of the treatment groups after 3% coumarin treatment (Table 1). Additionally, the survival rate significantly decreased with an increase in the concentration of coumarin (Figure 1B,C). These results indicated that coumarin significantly inhibited *S. litura* larval growth.

### 2.2. Transcriptome Analysis of S. litura after Exposure to Coumarin

After removing the reductant reads, a total of 43,661,616 (92.03%), 50,255,686 (91.59%), and 42,285,760 (92.77%) clean reads from the coumarin treatment at 0 h were obtained; 44,352,000 (92.68%), 42,673,846 (93.31%), and 42,683,020 (93.07%) clean reads from the coumarin treatment at 24 h were obtained; and 44,551,274 (94.61%), 42,626,972 (93.91%), and 42,774,280 (93.61%) clean reads from the coumarin treatment at 48 h were obtained (Appendix A). The values of Q20 (those with a base quality greater than 20) and Q30 (those with a base quality greater than 30) were approximately 96% and 91%, respectively. The values of the GC ontent in different samples were about 47% (Appendix A). Furthermore, 39,306,176 (90.02%), 45,643,668 (90.82%), and 38,038,940 (89.96%) clean reads from coumarin treatment groups at 0 h; 38,789,705 (87.46%), 37,983,568 (89.01%), and 38,137,885 (89.35%) clean reads from coumarin treatment groups at 24 h; and 39,942,407 (89.65%), 38,252,591 (89.74%), and 38,518,422 (90.05%) clean reads from coumarin treatment groups at 48 h were successfully mapped to the *S. litura* genome (Appendix A).

Based on the DESeq method, the DEGs were identified from different comparative groups. A total of 2906 DEGs were identified in Cou-24 h (coumarin treatment at 24 h) groups compared with the Cou-0 h (coumarin treatment at 0 h), of which 1487 DEGs were upregulated and 1419 DEGs were downregulated. A total of 1492 DEGs were identified in Cou-48 h (coumarin treatment at 48 h) groups compared with the Cou-24 h, of which 680 DEGs were upregulated and 812 DEGs were downregulated (Appendix A). Gene ontology (GO) enrichment analysis revealed that most DEGs were mainly associated with peptidase activity, serine hydrolase activity, and serine-type peptidase activity in Cou-24 h compared with Cou-0 h. By comparing Cou-48 h with Cou-24 h groups, most DEGs were mainly related to transmembrane transporter activity, transporter activity, and transmembrane transport (Appendix A). Additionally, Kyoto Encyclopedia of Genes and Genomes Pathway database (KEGG) enrichment analysis suggested that most DEGs were mainly involved in oxidative phosphorylation, carbon metabolism, citrate cycle (TCA cycle), and fatty acid metabolism at 24 h after coumarin treatment. Comparison of Cou-48 h with Cou-24 h groups showed that most DEGs were mainly involved in oxidative phosphorylation, lysosome, glutathione metabolism, and citrate cycle (TCA cycle) (Figure 2).

### 2.3. Identification of DEGs Related to Detoxification after Exposure to Coumarin

Transcriptome analysis showed that many DEGs related to detoxification were altered in different comparable groups. In total, 28 DEGs related to carboxylesterase were identified between Cou-24 h groups and Cou-0 h groups, among which 22 DEGs were upregulated from 0 h to 24 h after coumarin treatment and 6 DEGs were downregulated (Table 2). A total of 17 DEGs associated with glutathione *S*-transferase were identified, among which 5 DEGs were upregulated and 12 DEGs were downregulated from 0 h to 24 h after coumarin treatment (Table 2). A total of 35 DEGs related to cytochrome P450 were identified, of which 24 DEGs were upregulated from 0 h to 24 h after treated with coumarin and 11 DEGs were downregulated (Table 2). From 24 h to 48 h after coumarin treatment, a total of 20 DEGs related to carboxylesterase were screened, of which 17 DEGs were upregulated and three DEGs were downregulated (Table 3). In addition, a total of nine DEGs associated with GST were identified, of which seven DEGs were upregulated from 24 h to 48 h after exposure to coumarin, and the remaining two DEGs were downregulated (Table 3). A total of sixteen DEGs related to cytochrome P450 were identified from 24 h to 48 h after treatment with coumarin, of which five DEGs were upregulated and eleven DEGs were downregulated (Table 3).

Venn diagram analysis revealed that a total of 12 genes related to CarE showed differential expression in both Cou-24_vs_Cou-0 and Cou-48_vs_Cou-24 (Figure 3A). A total of 10 genes related to P450 exhibited difference in both Cou-24_vs_Cou-0 and Cou-48_vs_Cou-24 (Figure 3A). Additionally, a total of five genes associated with GST showed differential expression in both Cou-24_vs_Cou-0 and Cou-48_vs_Cou-24 (Figure 3A). Hierarchical cluster analysis also corresponded to expression patterns at 0, 24, and 48 h after coumarin treatment (Figure 3B and Appendix A).

To further validate the expression levels of DEGs involved in detoxification, a total of 12 genes were selected for RT-qPCR analysis. The results showed that the expression patterns of these genes can be divided into five categories. The first category involved DEGs that were consistently upregulated from 0 to 48 h after coumarin treatment, including *Venom-COE*, *GST-like*, *GST-1*, and *CYP324A16* (Figure 4). The second category included DEGs in which expression levels were downregulated from 0 h to 24 h after treatment with coumarin, while they were upregulated from 24 to 48 h, including *Cholinesterase 1*, *Esterase B1*, *AChE1*, and *P4506B6* (Figure 4). The third category included *P4509e2* and *P450304a1* in which expression levels were upregulated form 0 to 24 h after coumarin treatment (Figure 4). The expression level of *LOC111354038* showed no significant change from 0 to 24 h after coumarin treatment, whereas it was significantly upregulated from 24 to 48 h (Figure 4). These results indicated that coumarin treatment significantly affected the expression level of detoxification enzyme-related genes in *S. litura*. Notably, we found that the expression level of *SlCYP324A16* at 24 h after treatment with coumarin was 78.6 times that of 0 h, and that at 48 h after treatment was 224 times that of 0 h. We speculated that *SlCYP324A16* might play an important role in the detoxification of coumarin.

### 2.4. Determination of Detoxification Enzyme Activity

The activities of P450, GST, and AChE were measured to evaluate the effect of coumarin on detoxification enzyme activity. The results showed that CYP450 and AChE activities significantly decreased from 24 to 48 h after coumarin treatment, while they showed no significant change from 0 to 24 h (Figure 5. In contrast, GST activity was significantly increased from 0 to 24 h after treatment with coumarin, while it showed no significant change from 24 to 48 h (Figure 5). These results indicated that coumarin significantly reduced CYP450 and AChE activity and increased GST activity.

### 2.5. Effect of Coumarin on S. litura after Silencing of SlCYP324A16

RT-qPCR analysis revealed that *SlCYP324A16* gene expression increased rapidly from 0 h to 48 h after coumarin treatment. We considered that *SlCYP324A16* is an important target gene involved in the response to coumarin. RNAi was performed to assess the effect of coumarin on *S. litura* after *SlCYP324A16* silencing. The results showed that *SlCYP324A16* expression level was significantly downregulated at 48 h after dsRNA injection (Figure 6B). Furthermore, *SlCYP324A16* silencing resulted in a significant increase in the cumulative mortality compared with ds*GFP* groups (Figure 6C). Additionally, the weight of *S. litura* larvae significantly reduced in both ds*GFP* groups and ds*SlCYP324A16* groups after coumarin treatment. *S. litura* larval weight has no obvious difference between ds*GFP* and ds*SlCYP324A16* after H_2_O treatment, whereas larval weight in ds*SlCYP324A16* group was lower than ds*GFP* group after coumarin treatment (Figure 6D). These results indicated that the knockdown of the *SlCYP324A16* gene significantly increased the sensitivity of *S. litura* to coumarin.

### 2.6. Identification of DEGs Involved in Glycolysis and TCA cycle after Exposure to Coumarin

The transcriptome analysis revealed that several DEGs involved in glycolysis and TCA cycle were modified in different comparable groups. A total of 25 DEGs involved in glycolysis were identified and showed upregulation from 0 to 24 h after coumarin treatment. In addition, a total of 23 DEGs involved in the TCA cycle were identified between Cou-24 h groups and Cou-0 h groups, among which 22 DEGs were downregulated from 0 h to 24 h after coumarin treatment and one gene was upregulated (Table 4). From 24 to 48 h after coumarin treatment, a total of four glycolysis-related DEGs were identified, and these genes were upregulated compared Cou-48 h groups and Cou-24 h groups. A total of 12 DEGs associated with TCA cycle were identified and showed upregulation from 24 to 48 h after coumarin treatment (Table 5). KEGG enrichment analysis also revealed that the expression levels of key genes involved in glycolysis were significantly downregulated from 0 to 24 h after treatment with coumarin. However, glycolysis-related genes had no significant change from 24 to 48 h after coumarin treatment (Figure 7). Additionally, TCA cycle-related genes were also significantly downregulated at 24 h after coumarin treatment, while these genes were upregulated from 24 to 48 h after coumarin treatment (Figure 8). These results indicated that coumarin significantly inhibited *S. litura* glycolysis and TCA cycle at 24 h, while these pathways were activated at 48 h.

### 2.7. Metabolite Profiling of S. litura Hemolymph after Exposure to Coumarin

Metabolic analysis identified a total of 4498 metabolites using UPLC-MS (2345 and 2153 for negative- and positive-ion modes, respectively) in *S. litura* hemolymph from coumarin treatment groups at different time points. Of these, 4219 were categorized into 10 classified super classes, including lipids and lipid-like molecules (1173), organic acids and derivatives (857), organoheterocyclic compounds (749), benzenoids (498), organic oxygen compounds (431), phenylpropanoids and polyketides (183), nucleosides, nucleotides, and analogues (136), organic nitrogen compounds (70), and others (122) (Appendix A). 

The good stability and repeatability of the analysis were revealed using PCA and PLS-DA score plots. The results showed that a clear separation between control groups and coumarin treatment groups at 0 h, 24 h, and 48 h were exhibited on score plots of PCA and PLS-DA, indicating that two coumarin treatment groups at 24 h and 48 h had a distinct metabolic profile, but the score plots at 0 h overlapped between control groups and coumarin treatment groups (Figure 9A,B). A seven-fold cross validation R2Y (0.999) and Q2 (0.973) of the comparison of the Cou-24 h groups with Cou-0 h groups, showed fitness and predictability, and the negative Q2 in the 200-times response permutation testing revealed no overfitting in OPLA-DA (Figure 9C). A seven-fold cross validation R2Y (0.996) and Q2 (0.945) of the comparison of the Cou-48 h groups with Cou-24 h groups, showed fitness and predictability, and the negative Q2 in the 200-times response permutation testing revealed no overfitting in OPLA-DA (Figure 9C).

### 2.8. Effect of Coumarin Treatment on Metabolite Profiling in S. litura Hemolymph

The metabolic profiles in *S. litura* hemolymph after coumarin treatment at 0 h, 24 h and 48 h were determined to assess the effect of coumarin on hemolymph. According to the criteria of VIP value > 1 in the OPLS-DA analysis and FDR (q-value) < 0.05 in the Benjamini-Hochberg method, a total of 391 significantly-expressed metabolites (220 upregulated and 171 downregulated) were identified from 0 h to 24 h after treatment with coumarin, and 352 significantly-expressed metabolites (211 upregulated and 141 downregulated) were identified from 24 h to 48 h (Figure 10A,C; Appendix A). Further analysis revealed that the metabolites identified from 0 h to 24 h could be grouped into 11 classified super classes, with organic acids and derivates (100) being the most altered compounds in terms of metabolite numbers (Appendix A). The second super classes in metabolite numbers were lipids and lipid-like molecules (70) and organoheterocyclic compounds (63) were identified in both super classes. From 24 h to 48 h after coumarin treatment, identified metabolites could be grouped into 11 classified super classes, of which organic acids and derivates (90) were the most altered compounds in terms of metabolite numbers, and the second super class in metabolite numbers were lipids and lipid-like molecules (68) and organoheterocyclic compounds (52) were identified in both super classes (Appendix A).

### 2.9. Metabolic Pathway of S. litura Affected by Coumarin

An analysis of KEGG enrichment was performed to understand the biological functions of identified metabolites after coumarin treatment at different time points The results showed that a total of 15 pathways were found from 0 to 24 h after coumarin treatment, among which the most enriched pathways were mainly associated with amino acid and nucleotide metabolism as well as energy metabolism, including purine metabolism, alanine, aspartate and glutamate metabolism, lysine metabolism and citrate cycle (Figure 11A; Appendix A). Correlation analysis revealed that a total of 14 differential metabolites showed positive correlation, including alpha-lactose, 9-hydroxy-7*E*-Nonene-3,5-diynoic acid, 2,2-dichloro-12-(4-chlorophenyl) dodecanoic acid, 2-hydroxyphenethylamine, 1-nitro-3,5-dinitroso-1,3,5-triazinane, acetamide, L-histidine, trans-cinnamic acid, *N*-(1-deoxy-1-fructosyl) tyrosine, L-alloisoleucine, Indoleacrylic acid, gluconic acid, flurtamone, and betamethasone phosphate (Figure 11B). From 24 h to 48 h after treatment with coumarin, a total of 17 pathways were detected, of which most were mainly related to ABC transporters, arginine biosynthesis, butanoate metabolism, and alanine, aspartate, and glutamate metabolism (Figure 11C). A total of 15 differential metabolites exhibited positive correlation, among which D-Malic acid, 2,2-dichloro-12-(4-chlorophenyl) dodecanoic acid, 1-nitro-3,5-dinitroso-1,3,5-triazinane, and pyrogallol-2-*O*-glucuronide (Figure 11D) had a strong positive correlation. Additionally, D-Proline and mandelonitrile rutinoside also exhibited positive correlation (Figure 11C).

## 3. Discussion

Pest management currently faces a worldwide ecological challenge resulting from environmental pollution and insecticide resistance caused by the extensive use of synthetic chemical insecticides [32]. Plant-derived pesticides are cheap, biodegradable, ecofriendly, and act by several mechanisms of action in a more specific way. In plants, secondary products are small molecular weight compounds produced by secondary metabolic pathways and exhibit superior insecticidal properties [33]. The secondary phenolic metabolites are generally regarded as defensive molecules that can disrupt insect growth and oviposition [34]. In the current study, we found that natural coumarin significantly inhibited the growth and development of *S. litura* larvae. An increase in the concentration of coumarin resulted in a significant decrease in the weight of *S. litura* larvae. Sharma et al. revealed that coumarin caused a high percentage of egg and larval mortality of the potato tuber moth, *Phthorimaea operculella* Zell [35]. Moreover, a total of four coumarins isolated from the crude seed extract of Mammea siamensis-surangin B, surangin C, mammea E/BB, and mammea E/BC were reported to have insecticidal properties [36]. These results indicated that coumarins are highly phytotoxic and effective in controlling pests. 

GO enrichment analysis showed that DEGs were mainly related to peptidase activity and transmembrane transporter activity in *S. litura* after coumarin treatment. Most of genes related to peptidase activity showed significant difference and indicated that coumarin treatment may increase the proteolysis at 24 h. However, DEGs associated with ABC transporters and metabolism of xenobiotics by cytochrome P450 exhibited obvious difference at 48 h. We considered that coumarin treatment significantly increased the expression levels of insecticide resistance-related genes. By transcriptome analysis, a total of 80 and 45 DEGs related to detoxification were identified from 0 to 24 h and 24 to 48 h after coumarin treatment, respectively. Among them, most of carboxylesterase-related genes were downregulated from 0 to 24 h, while they were upregulated from 24 to 48 h after coumarin treatment. Our results indicated that coumarin could significantly inhibit the carboxylesterase in the early stages of processing. In insects, carboxylesterase enzymes are frequently implicated in the resistance to organophosphorus, carbamates, and pyrethroids [37]. Pengsook et al. found that thymol isolated from *Alpinia galanga* could significantly decrease carboxylesterase in *S. litura* [38]. GSTs are phase II metabolizing isozymes that catalyze the reaction between reduced glutathione and conjugate xenobiotic compounds, which in particular contribute significantly to organophosphate, organochlorine, cyclodiene, and pyrethroid resistance [39]. We also found that the expression levels of DEGs associated with GST were significantly downregulated after coumarin treatment in *S. litura*. However, GST activity significantly increased at 48 h after coumarin treatment. Duan et al. found that citalopram and mirtazapine increased GST activity in Daphnia magna, indicating that psychoactive drugs posed a high toxic threat to aquatic organisms [40]. We considered that coumarin significantly inhibited the expression level of GST-related genes, and GST enzyme activities were increased to compensate for the deficiency of the enzyme. P450 plays a vital role in the metabolism of insecticides and plant allelochemicals in insects [41]. We found that *SlCYP324A16* was rapidly upregulated from 0 to 24 h after coumarin treatment. Silencing of *SlCYP324A16* by RNA interference significantly increased *S. litura* mortality and decreased larval weight. In previous research, the cytochrome P450 gene, *CYP321A1,* was significantly upregulated after exposure to tannin in *S. litura*’s midgut and body fat [41]. These results indicated that *SlCYP324A16* could be an important target in the response to coumarin. 

Comparative transcriptome analysis showed that DEGs in glycolysis pathway were downregulated from 0 to 24 h after coumarin treatment. In comparison, they had no significant change from 24 to 48 h. In addition, DEGs in the citrate cycle were downregulated from 0 to 24 h after treatment; however, their expression levels were significantly upregulated from 24 to 48 h. These results indicated that coumarin exposure reduced the energy-related metabolism of *S. litura* by inhibiting the expression of key genes in glycolysis and citrate cycle. In our previous research, NMR-based metabolomic analysis revealed that *S. litura* glycolysis and TCA cycle were significantly inhibited after exposure to validamycin [42]. Li et al. also found that sanguinarine impaired trehalose hydrolysis, reduced trehalase activity and transcription, and led to the inhibition of energy metabolism in *B. mori* [18]. Based on these results, we considered that both validamycin and sanguinarine disrupted the insect’s energy metabolism by inhibiting trehalase activity. However, it is not clear whether coumarin works by inhibiting trehalase activity. Glycolysis and TCA cycle are important sources of energy for insects and are crucial for their growth and development [43]. The results indicated that *S. litura* larvae entered a starvation state, which resulted in inadequate energy intake. 

PCA analysis of metabolic profiles suggested that the coumarin-treated group overlapped with control group at 0 h. However, the metabolic profiles of the coumarin-treated group could be entirely separated from that of the control group. At 48 h, the metabolic profiles of the treatment group and control group were less overlapped. These results further indicated that significant changes in the metabolic pattern in *S. litura* occurred, 0–24 h after ingestion of coumarin. We speculated that the coumarin had significant effects on physiological metabolism at the early stage (0–24 h) of midgut entry, while the effect weakened at the late stage (24–48 h). KEGG enrichment analysis showed that identified differential metabolites from 24 to 48 h after coumarin treatment were mainly related to ABC transporters. ABC transporters are widely distributed in organisms and play an important role in the transport of xenobiotics [44]. In *S. frugiperda*, camptothecin treatment significantly induced the upregulation of ABC transporter-related genes in the fat body. Plants can release a variety of toxic secondary metabolites to deter the feeding of herbivorous insects. As part of these countermeasures, members of ABC transporter family play a crucial role in overcoming multiple chemical plant defenses [44]. Therefore, we considered that ABC transporters may be involved in transporting coumarin extracellular during the stage of coumarin treatment, thereby reducing the toxicity of coumarin against *S. litura*.

## 4. Materials and Methods

### 4.1. Spodoptera litura Rearing and Toxicity Assay

The *S. litura* larvae were collected from the orange orchard at Gannan Normal University, Ganzhou, China. Larvae were reared for six generations using artificial diets containing wheat germ, yeast, carrageenan, konjac flour, sorbic acid, vitamin C, corn oil, and linoleic acid at 25 °C, 65% relative humidity, and a photoperiod of 12:12 h (L:D). The adults after molting were transferred into a clean plastic container and fed using 10% honey soaked in cotton balls. 

Toxicity assay was performed following a previous protocol with minor modifications [42]. A total of 240 third-instar larvae were randomly divided into 12 groups (n = 20 in each group) for the toxicity assay of coumarin. The coumarin (98%) was purchased from Aladdin Industrial Corporation (Shanghai, China). After weighing, three groups of larvae were supplemented with the same amounts of artificial diets containing 1% coumarin, 2% coumarin, and 3% coumarin, respectively. The control group was fed diet only. Each group had three biological replicates. The weight of individual larvae in each group and the number of dead larvae in different groups were recorded daily. The survival rates were calculated using the formula: survival rate = (the number of surviving larvae − the number of dead larvae)/20 × 100%. The phenotype of *S. litura* larvae were observed at the fifth day after coumarin treatment using a camera. 

### 4.2. Extraction of RNA, cDNA Synthesis, and RT-qPCR Analysis

The total RNA was extracted from different samples using the animal tissue total RNA kit (Simgen, Hangzhou, China). RNA quality and quantity were measured with a NanoDrop 2000 spectrophotometer (Thermo Fisher Scientific, New York, NY, USA). The total RNA was reverse-transcribed in a 20 µL reaction system using a cDNA synthesis master mix kit according to a previous protocol. All cDNA samples were diluted to the same concentration that served as a template for RT-qPCR analysis. The RT-qPCR reactions were performed using a LightCycler^®^96PCR Detection System (Roche, Basel, Switzerland). The specific sample addition procedures were performed following a previous report [45]. The relative expression levels were analyzed using the 2^−∆∆Ct^ method. The constitutively expressed *glyceraldehyde-3-phosphate dehydrogenase* (*GAPDH*) was used as a reference gene. All primers are presented in Appendix A. There were three biological and technical replicates conducted for each sample.

### 4.3. cDNA Library Preparation and Sequencing

A total of 180 fifth-instar larvae were divided into three groups (60 larvae in each group) and treated with 3% coumarin. The high quality of RNA from *S. litura* midgut in each group was used for cDNA library construction based on Illumina’s protocols, and cDNA libraries were constructed with the TruSeq RNA Sample Preparation Kit v2 (Illumina, San Diego, CA, USA). Then, the prepared libraries were evaluated on an Agilent BioAnalyzer 2100 system, followed by sequencing on an Illumina HiSeq platform at Novogene (Tianjin, China). The obtained raw reads in fastq format were further processed to remove contained N base and low quality of reads using in-house perl scripts. The Q30 (percentage of sequences with sequencing error rate lower than 0.1%), Q20 (percentage of sequences with sequencing error rate lower than 1%), and the GC content of the clean data were determined.

### 4.4. Data Analysis, DEGs Identification, and Bioinformatics Analysis

Based on sequencing results, the expression levels of the genes were calculated using Fragments Per Kilobase of transcript sequence per Millions (FPKM). Differential expression analyses of two groups were conducted using the DESeq2 (Version 1.24.0) software, and differentially expressed genes (DEGs) were identified by a threshold of |log2(fold change)| ≥ 1 and *p*-value < 0.05. Gene Ontology (GO) enrichment analysis of DEGs was implemented using the clusterProfiler R package (3.8.1). GO terms with corrected *p*-value < 0.05 were considered as significantly enriched by DEGs. A Kyoto Encyclopedia of Genes and Genomes (KEGG) pathway enrichment analysis for the DEGs was performed using KOBAS. A *p*-value of <0.01 was set as the threshold.

### 4.5. Determination of Various Enzymes Activity

A total of 90 third-instar larvae were divided into three groups (n = 30 in each group) for enzyme determination. The *S. litura* midgut samples were collected from 0 h, 24 h, and 48 h after 3% coumarin treatment. P450 enzyme activities from different samples were measured according to previous protocol with minor modification [46]. In brief, the *S. litura* midgut samples were ground into powder by liquid nitrogen, and homogenized in 1 mL Tris-HCl buffer (0.1 M, pH 8.6) by a glass homogenizer. The mixture was centrifuged at 10,000× *g* for 15 min at 4 °C. The obtained supernatant was collected as an enzyme solution. The above enzyme solution was added into 5 μL of 10 mM NADPH, and incubated for 1 min at 34 °C. Approximately 250 μL of 0.02 mM 7-ethoxycoumarin O-deethylase (ECOD) was added, and enzyme activity was measured using a fluorospectrophotometer. Standard curve was prepared from 10 μL of stepwise-diluted solution (0.05–0.5 mM) of 7-hydroxycoumarin.

The AChE enzyme activity was measured using a kit purchased from Nanjing Jiancheng Bioengineering Institute (Nanjing, China), following the manufacturer’s instructions. Briefly, moderate saline was added to different midgut samples, ground using an electric grinding machine, and centrifuged at 2500 rpm for 10 min. The supernatant from different samples were collected and kept for later use. Additionally, the protein concentration of different samples was measured using the BCA method. The reaction mixture was prepared by adding different volume of samples, 1 μmol/mL standard application solution, ddH_2_O, substrate buffer, and color application solution, and the mixture was placed at 37 °C for 6 min. Approximately 30 μL of inhibitor and 100 μL of transparent agent were added and placed at room temperature for 15 min. The absorbance was measured at 412 nm. The AChE content was calculated using the formula: AChE content = (A_measure_ − A_control_)/(A_standard_ − A*blank*) × C_standard_/C_pr_.

The GST enzyme activity was measured using a kit purchased from Nanjing Jiancheng Bioengineering Institute (Nanjing, China) following the manufacturer’s instructions. In brief, moderate saline was added to different midgut samples, ground using an electric grinding machine, and centrifuged at 2500 rpm for 10 min. The supernatant from different samples were collected and kept for later use. Additionally, the protein concentration of different samples was measured using the BCA method. The reaction mixture was prepared by adding different volume of matrix fluid and supernatant, and then incubated at 37 °C for 10 min. Furthermore, different volumes of liquid for application, ethanol, and supernatant were added and centrifuged at 4000 rpm for 10 min. The supernatant was collected for further color reaction. The color reaction system was prepared containing different volumes of GSH standard solvent application solution, 20 mol/L GSH standard buffer, supernatant, and liquid for application. The mixture was placed at room temperature for 15 min, and the absorbance was measured at 412 nm.

### 4.6. dsRNA Synthesis and Microinjection

*SlP4506k1* gene was silenced using RNAi to determine its role in the response of *S. litura* midgut to coumarin treatment. Double-stranded *SlCYP324A16* (ds*SlCYP324A16*) and green fluorescent protein (ds*GFP*) were synthesized using the T7 RioMAX Express RNAi System (Promega, San Luis Obispo, CA, USA) according to the manufacturer’s instructions. The specific primers containing a T7 promoter sequence were designed using the Primer Premier 5 software and presented in Appendix A. The synthetic ds*SlCYP324A16* was diluted to 800 ng/µL working solution using RNase-free water. For the RNAi experiment, a total of 180 fourth-instar *S. litura* larvae were divided into 12 groups. Among them, six groups were injected with ds*SlCYP324A16* (5 µL per larva). After 48 h, three groups were transferred to diets with 3% coumarin, and the remaining three groups were introduced to diets without coumarin. A similar amount of ds*GFP* was injected as a control. The control groups were treated with the same methods (Figure 6A). The efficiency of *SlCYP324A16* silencing was determined at 24 h and 48 h after injection of dsRNA using RT-qPCR. Furthermore, the cumulative mortality and average weight per larva were analyzed at 0, 24, 48, and 72 h in different groups. All data were analyzed using one-way ANOVA in the SPSS 16.0 software. P-values less than 0.05 and 0.01 were defined as significant and extremely significant, respectively.

### 4.7. Hemolymph Sample Pretreatment for UPLC-MS Analysis

For metabolomics analysis, a total of 360 fifth-instar larvae were divided into twelve groups, of which six groups were treated with 3% coumarin and the remaining six groups were treated with sterile water. The hemolymph was collected at 0 h, 24 h, and 48 h from treatment groups and control groups. For hemolymph sample pretreatment, a volume of 100 μL hemolymph was transferred to a 1.5 mL centrifuge tube, to which 300 μL of protein precipitator containing methanol and acetonitrile (Vmet:Vace = 2:1) and 2 μg/mL of 2-chlorophene alanine was added. The mixed samples were swirled for 1 min and further treated by ultrasound on ice for 10 min, and then placed at −40 °C for 30 min. The mixture was centrifuged at 13,000 rpm for 10 min at 4 °C. Approximately 200 μL of supernatant was transferred into a UPLC vial and dried. A total of 300 μL mixture of methanol and ddH_2_O (1:4) was added to dissolve the dried samples. The above samples were placed at −40 °C for 2 h, and further centrifuged at 13,000 rpm at 4 °C for 10 min. A total of 150 μL of supernatant was collected by using an injector and filtered through a 0.22 μm organic phase pinhole microfilter. Finally, the filtrate was transferred into a UPLC vial and stored at −80 °C for subsequent UPLC-MS analysis. Quality control (QC) samples were pooled by mixing all samples at an equal volume.

### 4.8. UPLC-MS Analysis for Untargeted Metabolite Profiling

UPLC-MS analysis was performed using an ACQUITY UPLC I-Class system (Waters Corporation, Milford, CT, USA) combined with VION IMS QTOF Mass spectrometer (Waters Corporation, Milford, CT, USA) to determine the metabolite profile from different samples. Approximately one microliter aliquot of the filtrate was injected into an ACQUITY UPLC BEH C18 column (1.7 μm, 2.1 × 100 mm; Waters Corp.) at a flow rate of 0.4 mL/min and a column oven at 45 °C. Water and acetonitrile/methanol (2/3, *v/v*) containing 0.1% formic acid were utilized as mobile phases A and B, respectively. Metabolite elution was conducted based on the following linear gradient: 0 min, 1% B; 1 min, 30% B; 2.5 min, 60% B; 6.5 min, 90% B; 8.5 min, 100% B; 10.7 min, 100% B; 10.8 min, 1% B; and 13 min, 1% B. All the samples were maintained at 4 °C.

Data acquisition was conducted in full-scan mode (*m/z* ranges from 50 to 1000) combined with MSE mode, including two independent scans with different collision energies (CE) were alternatively acquired during the run. The parameters of the mass spectrometry were as follows: a low-energy scan (CE 4 eV) and a high-energy (CE ramp 20–45 eV) to fragment the ions. Argon (99.999%) was used as collision-induced dissociation gas; scan time: 0.2 s; interscan delay: 0.02 s; capillary voltage: 2.5 kV; cone voltage: 40 V; source temperature: 115 °C; desolvation gas temperature: 450 °C; and desolvation gas flow, 900 L/h. The QCs were injected at regular intervals (every nine samples) throughout the analytical run to assess repeatability of the data. All procedures were carried out in the laboratory of Shanghai Luming Biotechnology Co., Ltd. (Shanghai, China).

### 4.9. Data Preprocessing and Statistical Analysis

The original LC-MS data were processed using the software progenesis QI v2.3 (Nonlinear Dynamics, Newcastle, UK) for baseline filtering, peak picking, integration, retention time (RT) alignment, peak alignment, and normalization. The main parameters of 5 ppm precursor tolerance, 10 ppm product tolerance, and 5% product ion threshold were applied. Compound identification were based on precise mass-to-charge ratio (*m/z*), secondary fragments and isotopic distribution using the Human Metabolome Database (HMDB), Lipidmaps (V2.3), Metlin, EMDB, PMDB, and self-built databases to conduct qualitative analysis. 

The extracted data were then further processed by removing peaks with missing values (ion intensity = 0) in more than 50% of the group, by replacing zero value by half of the minimum value, and by screening according to the qualitative results of the compound. Compounds with resulting scores below 36 (out of 60) points were also deemed to be inaccurate and removed. A data matrix was obtained from the positive- and negative-ion data.

The matrix was imported into R to carry out a principal component analysis (PCA) to observe the overall distribution among the samples and the stability of the whole analysis process. Orthogonal Partial Least-Squares-Discriminant Analysis (QPLS-DA) and Partial Least-Squares-Discriminant Analysis (PLS-DA) were utilized to distinguish the metabolites that differ between groups. To prevent overfitting, seven-fold cross-validation and 200 response permutation testing (RPT) were used to evaluate the quality of the model.

Variable importance of projection (VIP) values obtained from the OPLS-DA model was used to rank the overall contribution of each variable to group discrimination. A two-tailed Student’s *t*-test was further used to verify whether the difference in metabolites groups were significant. Differential metabolites were selected with VIP > 1.0 and *p*-value < 0.05. Additionally, the Kyoto Encyclopedia of Genes and Genomes (KEGG; http://www.genome.jp/kegg/, accessed on 1 April 2023) was used to identify important pathways.

## 5. Conclusions

Considering these results, we proposed a hypothetical diagram illustrating that AChE and CYP450 activities were inhibited after coumarin entered the *S. litura* midgut, which resulted in the disruption of the detoxification of coumarin (Figure 12). Furthermore, in the early stage of coumarin treatment, coumarin significantly inhibited glycolysis and TCA cycle pathways, resulting in a reduction in energy production. However, coumarin treatment in the late stage resulted in a high level of coumarin being transported from the intracellular to extracellular spaces by ABC transporters, contributing to the reduction in toxicity. Our results provide insights for elucidating the toxicity mechanism of coumarin and lay a foundation for controlling *S. litura*.

## Figures and Tables

**Figure 1 ijms-24-13177-f001:**
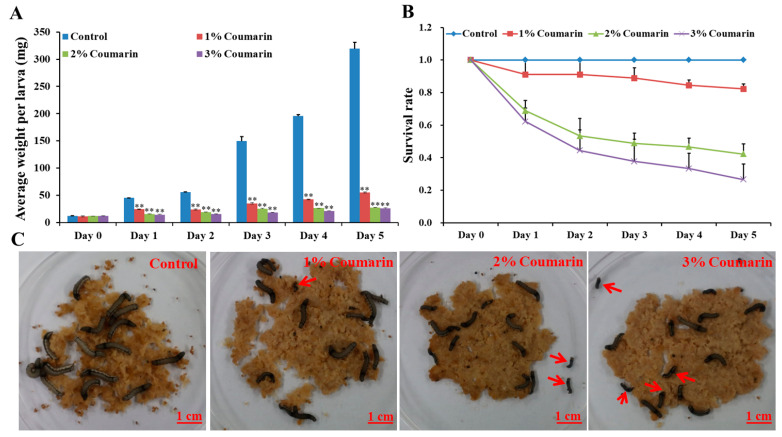
Effect of different concentration of coumarin on larval growth and development of *S. litura* from day 0 to day 5. (**A**) Analysis of average weight per larva after exposure to different concentration of coumarin from day 0 to day 5, Significant differences are indicated by ** *p* < 0.01; (**B**) Analysis of survival rate after exposure to different concentration of coumarin from day 0 to day 5; (**C**) Macroscopic phenotype of day-5 *S. litura* larvae treated with coumarin. The red arrow indicates the dead larvae.

**Figure 2 ijms-24-13177-f002:**
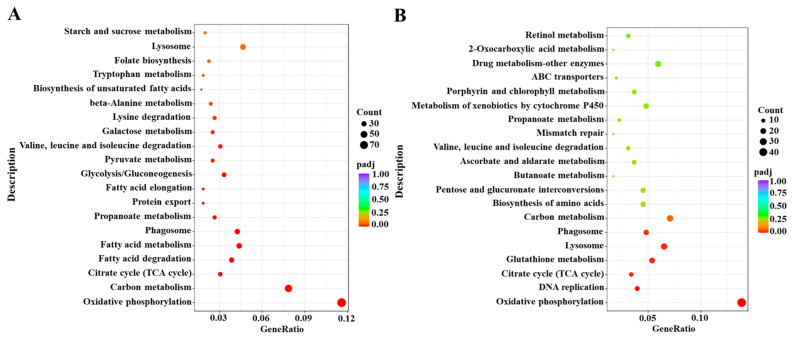
KEGG enrichment analysis of DEGs from 0 h to 24 h and 24 h to 48 h in *S. litura* after coumarin treatment. (**A**) KEGG enrichment analysis of DEGs from 0 h to 24 h in *S. litura* after coumarin treatment; (**B**) KEGG enrichment analysis of DEGs from 24 h to 48 h in *S. litura* after coumarin treatment. The sizes of the bubble indicate the number of DEGs enriched to the corresponding term. The color of the bubble indicates the Q value.

**Figure 3 ijms-24-13177-f003:**
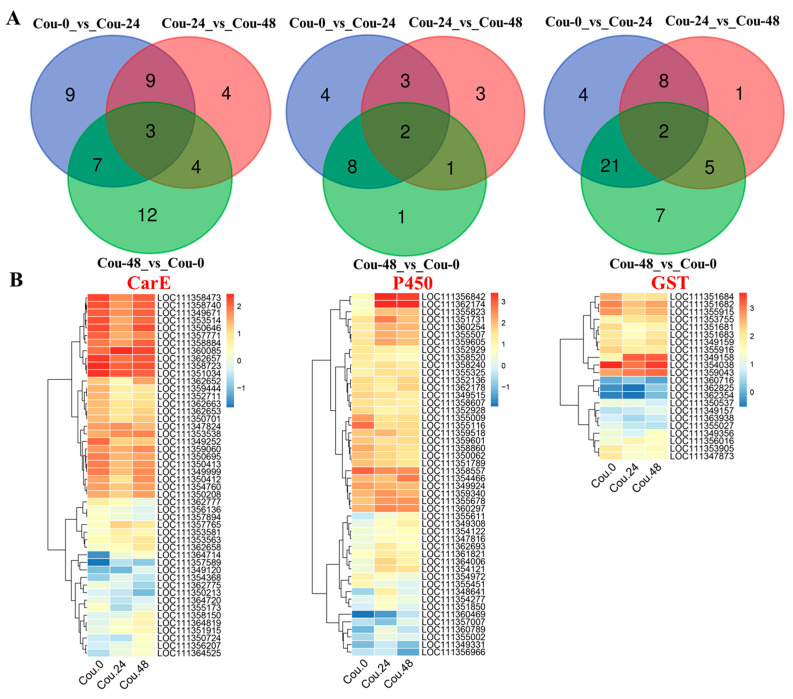
Venn diagram and clustering analysis of DEGs associated with CarE, P450, and GST in different comparative groups. (**A**) Venn diagram analysis of DEGs related to CarE, P450, and GST in Cou-0_vs_Cou-24, Cou-24_vs_Cou-48, and Cou-0_vs_Cou-48; (**B**) The clustering analysis of expression levels of DEGs at 0 h, 24 h, and 48 h after coumarin treatment.

**Figure 4 ijms-24-13177-f004:**
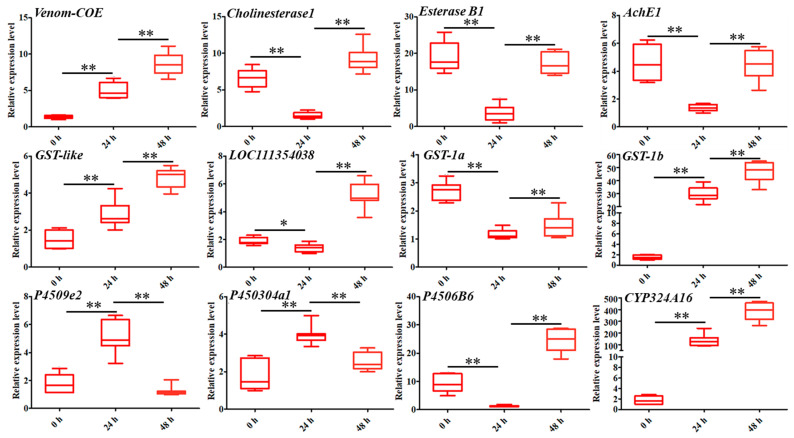
Analysis of the expression levels of twelve DEGs associated with CarE, GST, and P450 at 0 h, 24 h, and 48 after coumarin treatment, respectively. Data were normalized using *glyceraldehyde-3-phosphate dehydrase* (*GAPDH*) and are represented as the means ± standard errors of the means from three independent experiments. Significant differences are indicated by * *p* < 0.05 and ** *p* < 0.01.

**Figure 5 ijms-24-13177-f005:**
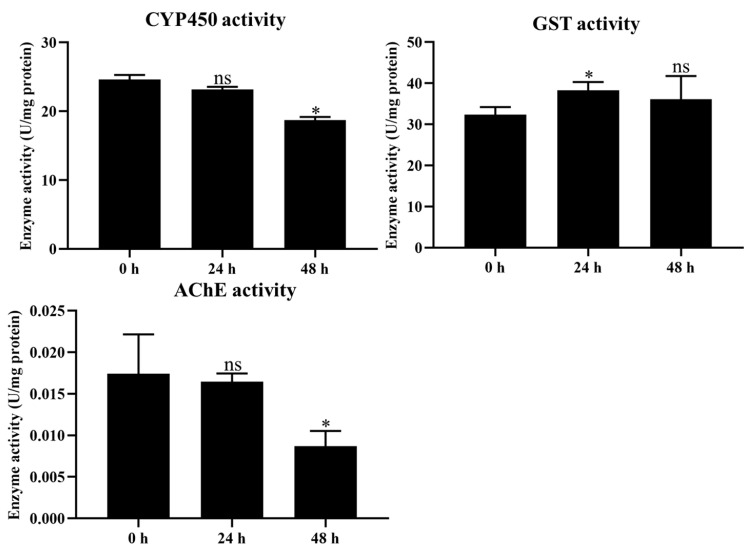
Measurement of CYP450, GST, and AChE activities at 0 h, 24 h, and 48 h after coumarin treatment. Each experiment contained three biological replicates. Statistical analysis was conducted using the SPSS 16.0 software. Significant differences are indicated by * *p* < 0.05. The ns indicates no significant difference.

**Figure 6 ijms-24-13177-f006:**
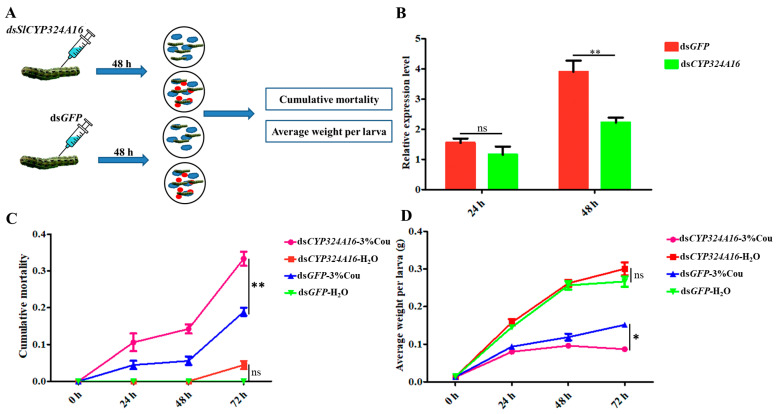
Effects of coumarin on the growth and development of *S. litura* after silencing of *SlCYP324A16* gene. (**A**) Illustration of the protocol used for RNA interference. The blue cloud symbol represented the artificial diet. The red oval symbol represented 3% coumarin; (**B**) Analysis of *SlP4506k1* expression levels after treatment with ds*SlP4506k1* and ds*GFP*; (**C**) Analysis of the cumulative mortality of *S. litura* larvae exposed to coumarin after silencing of the *SlP4506k1* gene; (**D**) Average weight of *S. litura* larvae exposure to coumarin after silencing of the *SlP4506k1* gene. Statistical analysis was conducted using the SPSS 16.0 software. The significant differences are indicated by * *p* < 0.05 and and ** *p* < 0.01. The ns indicates no significant difference.

**Figure 7 ijms-24-13177-f007:**
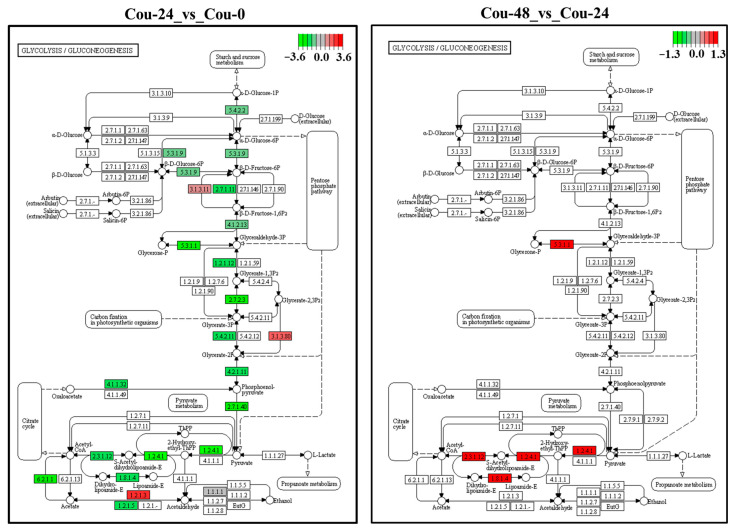
KEGG analysis of the changes in the glycolysis/gluconeogenesis pathway in *S. litura* at 0 h, 24 h, and 48 h after coumarin treatment. The red box indicates upregulation of DEGs and the green box indicates downregulation of DEGs.

**Figure 8 ijms-24-13177-f008:**
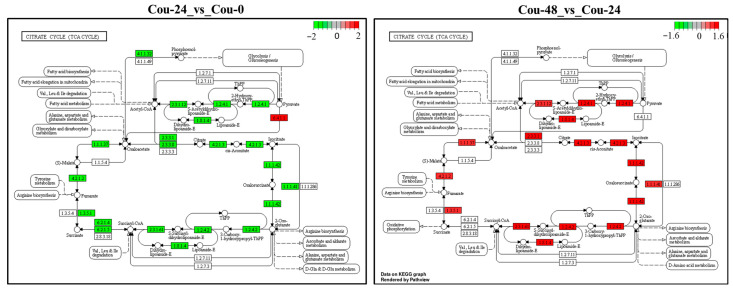
KEGG analysis of the changes in the TCA cycle pathway in *S. litura* at 0 h, 24 h and 48 h after coumarin treatment. The red box indicates upregulation of DEGs and the green box indicates downregulation of DEGs.

**Figure 9 ijms-24-13177-f009:**
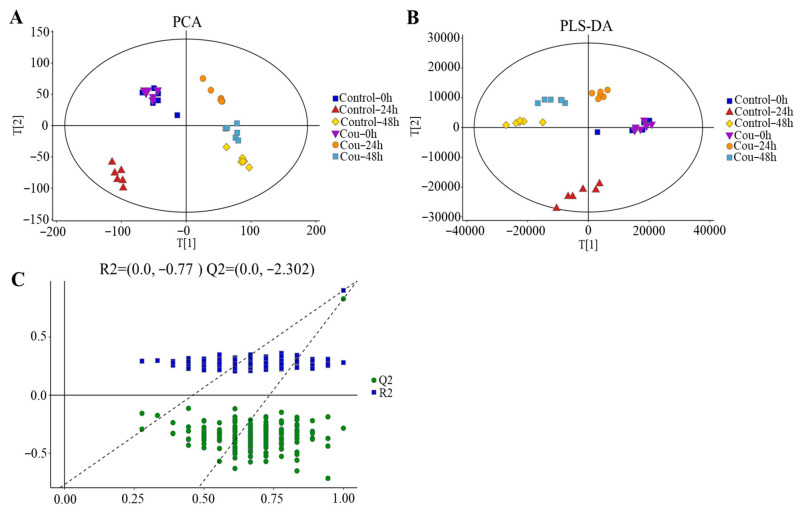
Score plots of multivariate analysis. (**A**) Principal component analysis (PCA) score plots for all samples. (**B**) Partial least squares-discriminant analysis (PLS-DA) score plots for different samples. (**C**) Results of 200-times response permutation testing of OPLS-DA. Q2 and R2 represent the intercepts of the regression curve and y-axis generated by the linear regression between the R2 and Q2 values of “permuted” model and the R2Y and Q2Y values. Control-0 h, Control-24 h, and Control-48 h: *S. litura* hemolymph samples were collected at 0 h, 24 h, and 48 h from control groups without coumarin; Cou-0 h, Cou-24 h, and Cou-48 h: *S. litura* hemolymph samples were collected at 0 h, 24 h, and 48 h from control groups containing coumarin.

**Figure 10 ijms-24-13177-f010:**
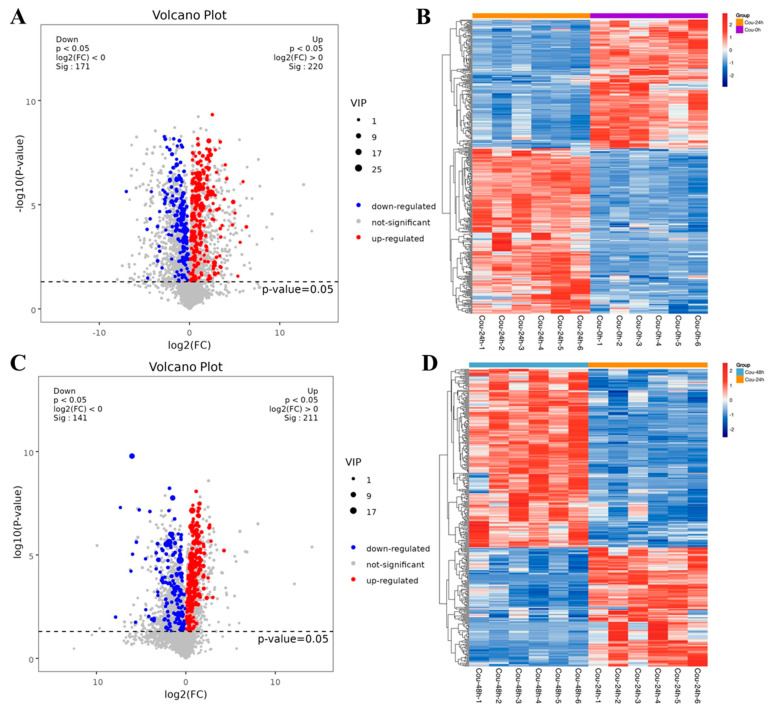
Expression levels of metabolites in different comparative groups. (**A**,**C**) Volcano plot for all differential metabolites in different comparative groups. Each dot indicates one metabolite with detectable expression in both conditions, with the colored dots marking the threshold (false discovery rate (FDR) < 0.05) for defining a metabolite as differentially expressed. Red and blue points indicate the significantly upregulated and downregulated metabolites, respectively; The gray points represent no significantly differential metabolites. (**B**,**D**) Hierarchical cluster analysis of all differential metabolites (FDR < 0.05). Each sample is represented as a single column and each metabolite is represented by a single row. Red coloration indicates significantly increased metabolite levels, and blue coloration indicates low metabolite levels.

**Figure 11 ijms-24-13177-f011:**
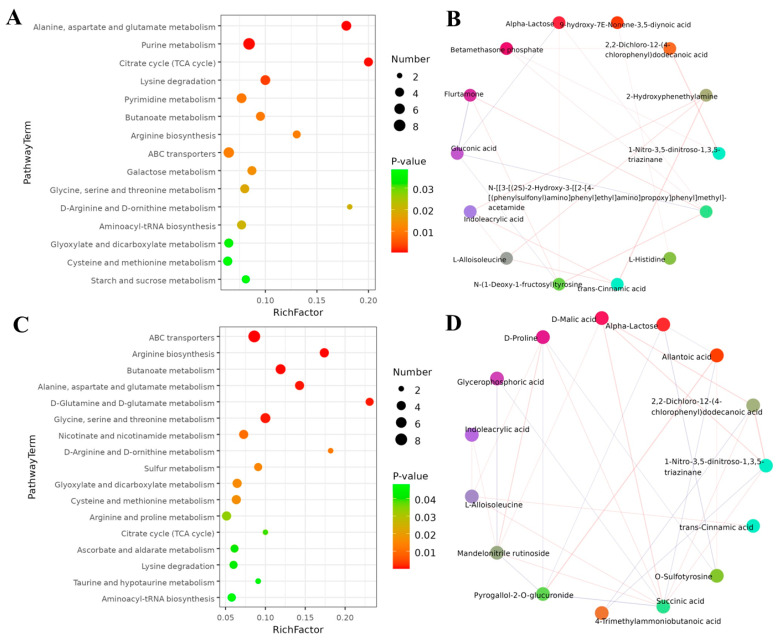
KEGG pathway enrichment analysis of differential metabolism after exposure to coumarin. (**A**,**C**) Significantly enriched pathways with FDR (q-value) < 0.05; (**B**,**D**) Relationships between different enriched metabolites.

**Figure 12 ijms-24-13177-f012:**
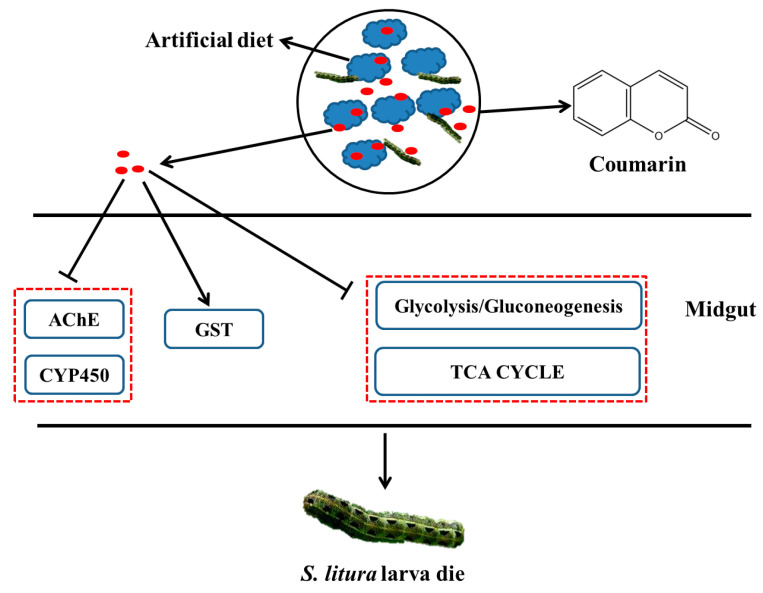
A hypothesized schematic diagram of coumarin effect on enzymes and glycometabolism in *S. litura*.

**Table 1 ijms-24-13177-t001:** Stagnant development in coumarin groups of *Spodoptera litura* from day 1 to day 5.

Group	Day 0 (mg)	Day 1 (mg)	Day 2 (mg)	Day 3 (mg)	Day 4 (mg)	Day 5 (mg)
Control-1	12.01 ± 2.39	45.51 ± 2.76	55.67 ± 2.71	151.90 ± 19.26	198.49 ± 31.54	304.45 ± 30.17
Control-2	12.58 ± 2.60	45.24 ± 2.87	55.50 ± 2.99	138.31 ± 23.43	196.13 ± 31.93	332.58 ± 30.08
Control-3	12.75 ± 2.00	45.35 ± 3.23	56.63 ± 2.46	158.86 ± 18.08	192.54 ± 22.92	320.94 ± 38.24
1% Coumarin-1	12.40 ± 2.29	23.95 ± 23.63	24.40 ± 2.40	33.30 ± 2.88	43.03 ± 2.73	53.85 ± 3.04
1% Coumarin-2	10.00 ± 1.99	25.39 ± 34.23	23.32 ± 1.98	35.70 ± 2.64	42.54 ± 4.22	56.32 ± 3.48
1% Coumarin-3	12.40 ± 2.56	24.70 ± 35.25	24.95 ± 3.19	37.10 ± 2.66	42.19 ± 2.62	56.14 ± 2.74
2% Coumarin-1	11.90 ± 3.97	16.49 ± 22.20	18.93 ± 4.54	26.41 ± 2.77	26.27 ± 2.04	27.68 ± 2.11
2% Coumarin-2	12.10 ± 2.11	15.02 ± 26.24	18.98 ± 3.75	24.30 ± 3.06	26.88 ± 2.98	27.27 ± 1.19
2% Coumarin-3	12.20 ± 2.67	14.90 ± 24.83	19.36 ± 3.33	25.40 ± 2.47	26.55 ± 2.40	27.58 ± 1.78
3% Coumarin-1	11.30 ± 2.32	15.24 ± 31.08	15.15 ± 3.52	17.56 ± 2.33	21.52 ± 2.54	25.85 ± 2.93
3% Coumarin-2	12.40 ± 2.42	14.63 ± 24.84	15.94 ± 2.68	18.42 ± 2.55	21.32 ± 2.30	27.22 ± 1.98
3% Coumarin-3	12.20 ± 2.98	13.36 ± 20.69	16.15 ± 1.89	18.93 ± 2.22	21.43 ± 2.11	24.76 ± 0.92

**Table 2 ijms-24-13177-t002:** Identification of DEGs related to detoxification from 0 h to 24 h after coumarin treatment.

Gene Name	Gene Description	Cou-0	Cou-24	Log2FC	*p*-Value
*Carboxylesterase*					
LOC111364714	venom carboxylesterase-6	0.4509	17.4966	5.4027	0.0039
LOC111357589	uncharacterized LOC111357589	1.5355	21.0749	3.7957	0.0004
LOC111357765	esterase FE4	67.1184	518.7683	2.95397	6.31 × 10^−63^
LOC111354368	neuroligin-4	10.6960	53.0543	2.2710	0.0005
LOC111353581	esterase FE4	56.1103	173.585	1.6292	0.0041
LOC111360085	acetylcholinesterase	2656.034	7885.301	1.5698	6.38 × 10^−6^
LOC111350208	venom carboxylesterase-6	982.7963	469.8686	−1.0656	0.0144
LOC111362777	liver carboxylesterase B-1	160.2841	70.9571	−1.1795	0.0028
LOC111358723	cholinesterase 1	16,835.48	7140.08	−1.2375	0.0036
LOC111349671	esterase E4	3680.21	1548.673	−1.2490	0.0050
LOC111362663	esterase FE4	726.0367	278.5143	−1.3836	4.71 × 10^−5^
LOC111362657	bile salt-activated lipase	2306.892	870.5927	−1.4061	0.0050
LOC111350695	esterase FE4	1904.922	711.3719	−1.4215	0.0014
LOC111350701	bile salt-activated lipase	706.9501	239.1579	−1.5651	0.0084
LOC111362653	esterase FE4	436.2362	144.3666	−1.5973	0.0022
LOC111358473	juvenile hormone esterase	5104.886	1636.45	−1.6415	7.51 × 10^−5^
LOC111358740	cholinesterase 1	4253.409	1281.325	−1.7312	6.72 × 10^−5^
LOC111349999	esterase E4	2440.619	728.2988	−1.7452	0.0002
LOC111350413	cholinesterase 2	1890.97	560.4996	−1.7549	0.0009
LOC111357771	uncharacterized LOC111357771	7112.863	2000.605	−1.8301	0.0018
LOC111350646	acetylcholinesterase-like	4633.256	1161.321	−1.9966	0.0002
LOC111350412	cholinesterase 2	1018.842	239.2248	−2.0916	4.85 × 10^−5^
LOC111353514	bile salt-activated lipase	4112.892	874.7928	−2.2335	4.40 × 10^−6^
LOC111352711	esterase FE4	705.1868	135.7773	−2.3783	1.16 × 10^−5^
LOC111359444	cholinesterase 1	1651.381	309.4374	−2.4160	1.08 × 10^−9^
LOC111362652	acetylcholinesterase-like	431.1727	75.45582	−2.5187	1.34 × 10^−5^
LOC111349252	esterase B1	6211.798	888.4212	−2.8059	1.17 × 10^−11^
LOC111355173	para-nitrobenzyl esterase	62.3453	6.9451	−3.1797	3.31 × 10^−7^
*Glutathione S-transferase*					
LOC111349158	glutathione S-transferase 1	695.0768	20,166.12	4.8588	1.16 × 10^−13^
LOC111356016	glutathione S-transferase-like	252.7765	743.835	1.5576	0.0003
LOC111353755	glutathione S-transferase 2	674.6568	1625.799	1.2701	0.0022
LOC111359043	pyrimidodiazepine synthase	8507.134	19,130.03	1.1691	0.0116
LOC111360716	failed axon connections	74.3522	154.7961	1.0601	0.0071
LOC111363938	metaxin-1	111.9925	50.8929	−1.1422	0.0073
LOC111355027	glutathione S-transferase 1	165.9887	73.52613	−1.1773	0.0005
LOC111349356	uncharacterized LOC111349356	406.9973	159.2776	−1.3556	0.0139
LOC111355916	glutathione S-transferase 1	3748.005	1440.041	−1.3801	0.0013
LOC111355915	glutathione S-transferase 1	9381.018	3157.125	−1.5712	0.0023
LOC111351682	glutathione S-transferase 1	10,388.87	3375.88	−1.6218	0.00078
LOC111354038	uncharacterized LOC111354038	75,761.86	22,874.34	−1.7277	5.15 × 10^−6^
LOC111351683	glutathione S-transferase 1	1588.893	449.4796	−1.8219	2.42 × 10^−6^
LOC111353905	glutathione S-transferase 2	1243.536	342.7152	−1.8601	1.45 × 10^−7^
LOC111351684	glutathione S-transferase 1	3947.215	1084.812	−1.8634	6.19 × 10^−6^
LOC111347873	glutathione S-transferase 1	1314.675	360.5861	−1.8668	0.0023
LOC111351681	glutathione S-transferase 1	3145.222	580.4061	−2.4382	2.22 × 10^−11^
*Cytochrome P450*					
LOC111362174	cytochrome P450 6k1-like	105.8283	69,473.7	9.3599	3.08 × 10^−46^
LOC111356842	cytochrome P450 6B6	231.7913	81,754.59	8.4622	6.31 × 10^−63^
LOC111348641	cytochrome P450 6B1	5.862026	407.2171	6.1330	0.0031
LOC111360789	probable cytochrome P450 305a1	1.954246	47.1659	4.6182	1.87 × 10^−6^
LOC111364006	cytochrome P450 9e2	26.28453	572.3499	4.4647	3.12 × 10^−15^
LOC111354121	cytochrome P450 6B6	67.53937	1455.844	4.4314	7.08 × 10^−29^
LOC111355823	cytochrome P450 6B5	190.4009	2531.378	3.7308	9.72 × 10^−10^
LOC111351731	cytochrome P450 9e2	1254.484	15,917.49	3.6659	2.41 × 10^−13^
LOC111362693	cytochrome P450 6B5	64.3353	629.8111	3.2822	3.54 × 10^−17^
LOC111351850	cytochrome P450 18a1	23.85004	175.475	2.8879	1.85 × 10^−6^
LOC111360254	cytochrome P450 6B2	578.855	3923.949	2.7610	4.98 × 10^−5^
LOC111349331	uncharacterized LOC111349331	4.44703	26.796	2.6392	0.0009
LOC111355611	cytochrome P450 6B6	12.77467	77.1950	2.5895	1.00 × 10^−5^
LOC111355507	probable cytochrome P450 6a13	924.8003	5556.457	2.5868	1.51 × 10^−10^
LOC111359605	uncharacterized LOC111359605	2481.81	14,894.9	2.5854	1.94 × 10^−7^
LOC111355002	cytochrome P450 6B2	19.17594	112.3051	2.5569	1.35 × 10^−7^
LOC111354277	probable cytochrome P450 301a1	40.4871	153.1612	1.9281	0.0018
LOC111361821	ecdysone 20-monooxygenase	204.7222	726.7638	1.8277	1.11 × 10^−9^
LOC111355325	cytochrome P450 6B2	1609.141	5636.928	1.8086	5.04 × 10^−8^
LOC111360297	cytochrome P450 6B5	3525.977	10,898.34	1.6280	4.65 × 10^−5^
LOC111347816	probable cytochrome P450 49a1	119.0582	352.8756	1.5677	3.83 × 10^−6^
LOC111355678	probable cytochrome P450 6a13	2675.074	7503.362	1.4879	0.0103
LOC111359340	cytochrome P450 9e2	3180.774	8024.88	1.3352	0.0049
LOC111358240	cytochrome P450 CYP12A2	536.533	1261.287	1.2334	0.0018
LOC111352136	NADPH–cytochrome P450 reductase	3667.327	1772.239	−1.0494	0.0137
LOC111358520	cytochrome P450 4d2	602.4715	277.1223	−1.1202	0.0036
LOC111358557	cytochrome P450 4d2	23,612.04	9980.55	−1.2423	0.0108
LOC111359601	uncharacterized LOC111359601	2497.045	6350.117	−1.3466	0.0002
LOC111354972	cytochrome P450 4C1	373.917	144.394	−1.3743	0.0084
LOC111362178	cytochrome P450 6k1	1018.111	365.548	−1.4785	0.0154
LOC111358860	cytochrome P450 6B2	5049.159	1726.827	−1.5481	0.0009
LOC111352928	cytochrome P450 4g1	2367.485	636.8526	−1.8945	1.65 × 10^−6^
LOC111355451	probable cytochrome P450 304a1	354.1661	83.6607	−2.0858	1.33 × 10^−9^
LOC111355009	cytochrome P450 4C1	7093.722	914.6282	−2.9552	2.16 × 10^−13^
LOC111355116	cytochrome P450 6B6	18,808.81	1298.406	−3.8567	9.99 × 10^−10^

**Table 3 ijms-24-13177-t003:** Identification of DEGs related to detoxification from 24 h to 48 h after coumarin treatment.

Gene Name	Gene Description	Cou-24	Cou-48	Log2FC	*p*-Value
*Carboxylesterase*					
LOC111362652	acetylcholinesterase-like	86.9740	1124.752	3.6972	2.70 × 10^−10^
LOC111362658	bile salt-activated lipase-like	88.1408	289.8123	1.7178	5.45 × 10^−10^
LOC111350412	cholinesterase 2-like	275.3892	1693.132	2.6216	6.09 × 10^−7^
LOC111358884	esterase FE4-like	1335.792	4039.308	1.5969	1.04 × 10^−6^
LOC111358150	juvenile hormone esterase-like	25.4487	123.1515	2.2870	9.77 × 10^−6^
LOC111349120	juvenile hormone esterase-like	4.1534	32.2946	2.9614	1.35 × 10^−5^
LOC111359060	acyl-protein thioesterase 2	695.2824	1401.097	1.0116	4.96 × 10^−5^
LOC111350724	cholinesterase 2-like	8.3791	66.7068	3.0232	0.0001
LOC111350646	acetylcholinesterase-like	1337.498	5676.586	2.0858	0.0002
LOC111358740	cholinesterase 1	1472.923	3836.439	1.3814	0.0008
LOC111353514	bile salt-activated lipase-like	1005.366	3377.045	1.7483	0.0010
LOC111350208	venom carboxylesterase-6	540.4641	1439.223	1.4139	0.0014
LOC111349999	esterase E4	836.7409	2310.262	1.4657	0.001493
LOC111347824	venom carboxylesterase-6	1761.006	778.9888	−1.1770	0.0033
LOC111355173	para-nitrobenzyl esterase	7.990478	30.3958	1.9470	0.003954
LOC111354368	neuroligin-4	60.4712	25.0773	−1.2749	0.0069
LOC111358473	juvenile hormone esterase	1879.844	3913.159	1.0580	0.0077
LOC111354760	juvenile hormone esterase-like	890.4963	2087.991	1.2300	0.0080
LOC111349671	esterase E4	1779.289	4029.061	1.1794	0.0091
LOC111362777	liver carboxylesterase B-1	81.2561	30.2480	−1.4280	0.0101
*Glutathione S-transferase*					
LOC111349356	uncharacterized LOC111349356	183.4426	975.2341	2.4125	2.57 × 10^−5^
LOC111350537	maleylacetoacetate isomerase 2	371.1178	178.4307	−1.0574	0.0013
LOC111351681	glutathione S-transferase 1	665.3634	1336.893	1.0075	0.0002
LOC111351683	glutathione S-transferase 1	515.4423	1607.81	1.6423	7.70 × 10^−8^
LOC111354038	uncharacterized LOC111354038	26,201.41	61,566.75	1.2325	0.0001
LOC111360716	failed axon connections	176.875	67.81142	−1.3852	9.73 × 10^−5^
LOC111362354	ganglioside-induced differentiation-associated protein 1	9.444761	46.72668	2.3120	0.0021
LOC111362825	ganglioside-induced differentiation-associated protein 1	9.6691	67.8948	2.8149	0.0019
LOC111349159	glutathione S-transferase 1	874.2036	2515.824	1.5254	0.0028
*Cytochrome P450*					
LOC111348641	cytochrome P450 6B1	464.3755	31.5317	−3.8818	1.40 × 10^−10^
LOC111354466	cytochrome P450 6B5	3093.616	17,813.34	2.5257	1.84 × 10^−8^
LOC111357007	cytochrome P450 4C1-like	3.2233	38.0246	3.5722	7.39 × 10^−7^
LOC111351850	cytochrome P450 18a1	199.1587	38.9521	−2.3552	7.20 × 10^−5^
LOC111356966	uncharacterized LOC111356966	33.9002	3.3296	−3.3401	0.0001
LOC111355611	cytochrome P450 6B6-like	88.6764	281.5663	1.6713	0.0003
LOC111349331	uncharacterized LOC111349331	30.4936	6.4089	−2.2582	0.0004
LOC111362178	cytochrome P450 6k1-like	421.0712	1851.439	2.1373	0.0007
LOC111349515	cytochrome P450 6B6	1377.623	662.0387	−1.0576	0.0015
LOC111362693	cytochrome P450 6B5	718.5333	183.6519	−1.9690	0.0016
LOC111351731	cytochrome P450 9e2	18,089.6	4118.491	−2.1350	0.0016
LOC111364006	cytochrome P450 9e2	650.4838	143.5995	−2.1804	0.0021
LOC111359518	cytochrome P450 9e2	3733.961	1712.212	−1.1250	0.0024
LOC111360469	cytochrome P450 4C1	1.07224	12.1195	3.6048	0.0028
LOC111352136	NADPH–cytochrome P450 reductase	2034.177	4703.494	1.2095	0.0037
LOC111360789	probable cytochrome P450 305a1	53.7626	12.2715	−2.1290	0.0047

**Table 4 ijms-24-13177-t004:** Identification of DEGs involved in glycolysis and TCA cycle from 0 h to 24 h after coumarin treatment.

Gene Name	Gene Description	Cou-0	Cou-24	Log2FC	*p*-Value
*Glycolysis/Gluconeogenesis*					
LOC111355527	pyruvate kinase-like	10,029.49	2008.17	−2.3204	3.80 × 10^−12^
LOC111360804	triosephosphate isomerase	3097.679	619.1088	−2.3234	2.42 × 10^−10^
LOC111347932	enolase	10,153.42	2413.292	−2.0730	2.11 × 10^−9^
LOC111356188	phosphoglycerate kinase	2259.451	461.2755	−2.2926	6.49 × 10^−9^
LOC111354384	probable pyruvate dehydrogenase E1	3054.369	813.3595	−1.9093	8.15 × 10^−8^
LOC111353812	dihydrolipoyllysine-residue acetyltransferase	4060.126	1149.014	−1.8214	1.73 × 10^−7^
LOC111360723	acetyl-coenzyme A synthetase	3195.666	552.6376	−2.5323	6.23 × 10^−7^
LOC111355302	alcohol dehydrogenase	2186.852	7120.034	1.7034	8.51 × 10^−7^
LOC111353761	glyceraldehyde-3-phosphate dehydrogenase	21,385.76	6552.904	−1.7065	1.27 × 10^−6^
LOC111359970	phosphoglycerate mutase 1	2525.376	707.2699	−1.8365	1.66 × 10^−6^
LOC111348281	aldehyde dehydrogenase	49,571.64	11,593.98	−2.0962	1.69 × 10^−6^
LOC111353109	ATP-dependent 6-phosphofructokinase	754.5358	236.7817	−1.6736	4.41 × 10^−6^
LOC111356157	pyruvate dehydrogenase E1	3664.081	1203.187	−1.6066	4.95 × 10^−6^
LOC111358748	alcohol dehydrogenase class-3	7719.941	2166.966	−1.8330	1.19 × 10^−5^
LOC111355208	fructose-1,6-bisphosphatase 1	622.5551	1580.132	1.3445	2.60 × 10^−5^
LOC111349213	dihydrolipoyl dehydrogenase	6660.689	2382.305	−1.4835	3.61 × 10^−5^
LOC111350235	aldehyde dehydrogenase	5779.774	1693.489	−1.7712	6.37 × 10^−5^
LOC111351185	glucose-6-phosphate isomerase	431.2507	192.815	−1.1634	0.0002
LOC111348104	phosphoglucomutase-2	1575.345	679.847	−1.2129	0.0005
LOC111350817	phosphoenolpyruvate carboxykinase	8196.958	2194.445	−1.9014	0.0006
LOC111359778	multiple inositol polyphosphate phosphatase 1	13.3643	55.7705	2.0892	0.0007
LOC111351708	putative aldehyde dehydrogenase	12909.5	6377.753	−1.0173	0.0008
LOC111360440	fructose-bisphosphate aldolase	8408.846	4001.305	−1.0715	0.0021
LOC111355564	retinal dehydrogenase 1	86.5984	1068.766	3.6245	0.0028
LOC111360439	fructose-bisphosphate aldolase	2010.322	773.8903	−1.3773	0.0060
*Citrate cycle (TCA cycle)*					
LOC111362068	pyruvate carboxylase	2486.594	8433.716	1.7623	5.34 × 10^−12^
LOC111351290	dihydrolipoyllysine-residue succinyltransferase	7572.191	1970.042	−1.9425	1.79 × 10^−9^
LOC111354947	probable isocitrate dehydrogenase	3068.617	925.75	−1.7293	4.72 × 10^−8^
LOC111354384	probable pyruvate dehydrogenase E1	3054.369	813.3595	−1.9093	8.15 × 10^−8^
LOC111348413	malate dehydrogenase	10839.8	2870.69	−1.9169	9.22 × 10^−8^
LOC111359296	aconitate hydratase	13267.16	3354.373	−1.9838	9.58 × 10^−8^
LOC111353812	dihydrolipoyllysine-residue acetyltransferase	4060.126	1149.014	−1.8214	1.73 × 10^−7^
LOC111356489	fumarate hydratase	2038.518	720.9951	−1.5001	3.09 × 10^−7^
LOC111359992	malate dehydrogenase	15,350.17	4136.996	−1.8917	6.61 × 10^−7^
LOC111357468	succinate dehydrogenase cytochrome b560 subunit	3175.645	872.7566	−1.8635	6.86 × 10^−7^
LOC111360629	probable citrate synthase	19,544.5	5754.938	−1.7640	2.94 × 10^−6^
LOC111356157	pyruvate dehydrogenase E1	3664.081	1203.187	−1.6066	4.95 × 10^−6^
LOC111351504	ATP-citrate synthase	2735.836	1023.545	−1.4187	5.31 × 10^−6^
LOC111350483	succinate dehydrogenase	5595.091	2138.591	−1.3875	2.97 × 10^−5^
LOC111349213	dihydrolipoyl dehydrogenase	6660.689	2382.305	−1.4835	3.61 × 10^−5^
LOC111353386	isocitrate dehydrogenase	10,257.99	3219.758	−1.6718	5.30 × 10^−5^
LOC111357097	isocitrate dehydrogenase	2194.843	935.3239	−1.2307	6.02 × 10^−5^
LOC111348703	succinate-CoA ligase	5418.423	2189.555	−1.3073	8.66 × 10^−5^
LOC111357155	succinate dehydrogenase	3485.447	1600.493	−1.1230	0.0003
LOC111350817	phosphoenolpyruvate carboxykinase	8196.958	2194.445	−1.9013	0.0006
LOC111349733	2-oxoglutarate dehydrogenase	7347.055	2748.649	−1.4185	0.0007
LOC111356570	fumarate hydratase	210.9027	70.2749	−1.5860	0.0009
LOC111353246	succinate-CoA ligase	3228.557	1464.208	−1.1411	0.0011

**Table 5 ijms-24-13177-t005:** Identification of DEGs involved in glycolysis and TCA cycle from 24 h to 48 h after coumarin treatment.

Gene Name	Gene Description	Cou-24	Cou-48	Log2FC	*p*-Value
*Glycolysis/Gluconeogenesis*					
LOC111354384	probable pyruvate dehydrogenase E1	933.7818	2266.246	1.2798	0.0002
LOC111360804	triosephosphate isomerase	710.7908	1714.832	1.2714	0.0002
LOC111349213	dihydrolipoyl dehydrogenase	2735.847	6653.498	1.2823	0.0002
LOC111353812	dihydrolipoyllysine-residue acetyltransferase	1319.113	2866.238	1.1200	0.0010
*Citrate cycle (TCA cycle)*					
LOC111351290	dihydrolipoyllysine-residue succinyltransferase	2258.894	5411.073	1.2606	3.14 × 10^−7^
LOC111359296	aconitate hydratase	3853.49	11,312.4	1.5538	9.75 × 10^−6^
LOC111353386	isocitrate dehydrogenase	3700.217	10,874.93	1.5555	7.44 × 10^−5^
LOC111356489	fumarate hydratase	826.2778	1681.034	1.0254	0.0001
LOC111354384	probable pyruvate dehydrogenase E1	933.7818	2266.246	1.2798	0.0002
LOC111349213	dihydrolipoyl dehydrogenase	2735.847	6653.498	1.2823	0.0002
LOC111354947	probable isocitrate dehydrogenase	1062.391	2351.336	1.1467	0.0003
LOC111357468	succinate dehydrogenase	1001.409	2087.434	1.0603	0.0004
LOC111360629	probable citrate synthase	6612.958	15,702.9	1.2478	0.0007
LOC111359992	malate dehydrogenase	4751.662	10,447.04	1.1367	0.0008
LOC111353812	dihydrolipoyllysine-residue acetyltransferase	1319.113	2866.238	1.1200	0.0010
LOC111349733	2-oxoglutarate dehydrogenase	3161.409	7903.156	1.3220	0.0021

## Data Availability

Data is contained within the article or Appendix A. For other information, please contact the corresponding author.

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
