# Peer review of "Natural Coumarin Shows Toxicity to Spodoptera litura by Inhibiting Detoxification Enzymes and Glycometabolism"

_ijms, 2023, doi:10.3390/ijms241713177_

Round 1

Reviewer 1 Report

The study on coumarin and its derivatives as potential insecticides, and their effect on Spodoptera litura larvae, is both fascinating and significant in the realm of pest control and agriculture. The researchers have meticulously explored the toxicological mechanisms of coumarin, revealing its potential as a natural and effective solution for insect pest management.

One of the most noteworthy aspects of this study is the use of natural coumarin to inhibit the growth and development of Spodoptera litura larvae. The toxicological assays conducted clearly demonstrate the potency of coumarin in inhibiting larval development, highlighting its potential as a valuable alternative to synthetic chemical insecticides.

The transcriptomic sequencing carried out to identify differentially-expressed genes (DEGs) related to detoxification after coumarin treatment provides valuable insights into the underlying molecular mechanisms. The discovery of 80 and 45 DEGs at different time intervals sheds light on the temporal dynamics of the response to coumarin exposure, further bolstering the scientific merit of this research.

The enzyme activity analysis conducted in the study is commendable, as it revealed crucial changes in CYP450, AChE, and GST activities after coumarin treatment. The observed decrease in CYP450 and AChE activities, along with the increase in GST activity, indicates that coumarin might disrupt vital detoxification processes in S. litura larvae. Such findings contribute significantly to understanding the mode of action of coumarin as an insecticidal agent.

The researchers' work in silencing the SlCYP324A16 gene through RNA interference to examine its effect on larval mortality and weight after coumarin treatment is another compelling aspect of this study. This experiment provided concrete evidence of the gene's involvement in the response to coumarin, strengthening the case for coumarin's potential use as an insecticide.

The exploration of gene expression levels related to glycolysis and the TCA cycle after coumarin treatment at different time points is an insightful addition to the research. The findings suggest that coumarin may have diverse effects on metabolic processes in S. litura larvae, underscoring the complexity of its mode of action.

The metabonomics analysis carried out to identify differential metabolites involved in purine metabolism, amino acid metabolism, TCA cycle, and ABC transporters following coumarin treatment further enhances the depth of this study. The comprehensive understanding of the impact of coumarin on these metabolites brings the researchers closer to unraveling the intricate interactions between coumarin and the insect's physiological processes.

In conclusion, this study significantly contributes to the field of insect pest management by providing compelling evidence of the insecticidal potential of natural coumarin and its derivatives. The integration of toxicological assays, transcriptomic sequencing, enzyme activity analysis, RNA interference, and metabonomics allows for a holistic view of the toxicological mechanism of coumarin on Spodoptera litura larvae. This research not only deepens our understanding of coumarin's mode of action but also opens doors for the development of eco-friendly and effective insecticides for sustainable agriculture. However, further research on the safety and efficacy of coumarin in field applications and its impact on non-target organisms would be beneficial before widespread adoption in agricultural practices. Overall, this study is a remarkable achievement in the realm of insecticidal research and warrants the attention of researchers and practitioners in the field.

The manuscript provides valuable insights into the toxicological mechanism of coumarin on Spodoptera litura larvae, particularly through the analysis of gene expression data. However, there are some aspects of the statistical analyses that could be improved to strengthen the reliability and interpretation of the results.

  1. Sample Size and Replicates: It is essential to provide information about the total number of samples used in the study and the allocation of these samples across the experimental groups. Additionally, details about the biological replicates (independent samples from different organisms) and technical replicates (repeated measurements from the same sample) should be clearly stated. Increasing the number of biological replicates can enhance the statistical power and improve the robustness of the findings.

  1. Statistical Test Selection: The manuscript mentions the use of DESeq2 for conducting differential expression analysis. While DESeq2 is a powerful tool for RNA-seq data, it is important to ensure that the assumptions of the statistical methods are met and that the appropriate test is chosen based on the experimental design. Considering the potential for batch effects or other sources of variability, it may be beneficial to explore other differential expression analysis tools to validate the findings.

  1. Multiple Testing Correction: The manuscript indicates the use of a P-value threshold of <0.05 for identifying differentially expressed genes (DEGs) and a P-value threshold of <0.01 for KEGG pathway enrichment analysis. However, it is crucial to address the issue of multiple testing to control for false positives. Applying appropriate multiple testing correction methods such as the Benjamini-Hochberg procedure for the DEG analysis and Bonferroni correction for the pathway enrichment analysis can help mitigate the risk of false discoveries.

  1. Effect Size Estimation: In addition to using a log2(fold change) threshold of ≥1 to identify DEGs, it is essential to report effect size estimates and confidence intervals. Effect size measures provide a more interpretable representation of the magnitude of gene expression changes and allow readers to better understand the biological significance of the results.

  1. Presentation of Results: The results of the differential expression analysis and pathway enrichment analysis should be presented in a clear and comprehensive manner. Providing volcano plots or heatmaps can be helpful in visualizing the distribution of DEGs and their functional enrichment.

  1. Biological Interpretation: While GO and KEGG analyses are valuable tools for functional interpretation, it is crucial to provide a clear biological interpretation of the enriched pathways and their relevance to the insecticidal properties of coumarin. Discussion of the potential implications of the identified DEGs and pathways on the toxicological mechanisms would enhance the impact and significance of the study.

In conclusion, the manuscript offers valuable insights into the toxicological mechanism of coumarin on S. litura larvae. However, improvements in the statistical analyses, particularly in sample size, statistical test selection, multiple testing correction, effect size estimation, and presentation of results, would strengthen the robustness and interpretation of the findings. Providing a clear biological context for the enriched pathways will enhance the overall impact and significance of the study. Addressing these points would elevate the quality and validity of the research, making it more valuable to the scientific community.

Author Response

  1. Sample Size and Replicates: It is essential to provide information about the total number of samples used in the study and the allocation of these samples across the experimental groups. Additionally, details about the biological replicates (independent samples from different organisms) and technical replicates (repeated measurements from the same sample) should be clearly stated. Increasing the number of biological replicates can enhance the statistical power and improve the robustness of the findings

Reply: Thanks for your thoughtful comments. In the current research, for toxicity assay, a total of 240 third-instar larvae were randomly divided into 12 groups (20 larvae in each group). Each of the three groups was treated with 1% coumarin, 2% coumarin, and 3% coumarin, respectively. The remaining three groups were used as the control. For transcriptome sequencing, a total of 180 fifth-instar larvae were divided into three groups (60 larvae in each group) and treated with 3% coumarin. The high quality of RNA from S. litura midgut in each group was used for cDNA library construction. For metabolomics analysis, a total of 360 fifth-instar larvae were divided into twelve groups, of which six groups were treated with 3% coumarin and the remaining six groups were treated with sterile water. The hemolymph was collected at 0 h, 24 h and 48 h from treatment groups and control groups. We have added the related descriptions in previous manuscript.

  1. Statistical Test Selection: The manuscript mentions the use of DESeq2 for conducting differential expression analysis. While DESeq2 is a powerful tool for RNA-seq data, it is important to ensure that the assumptions of the statistical methods are met and that the appropriate test is chosen based on the experimental design. Considering the potential for batch effects or other sources of variability, it may be beneficial to explore other differential expression analysis tools to validate the findings.

Reply: Thanks for your valuable comments. To date, three are three methods to conduct differential expression analysis. The first method is the “Trimmed Mean of M-values” normalization (TMM) and implemented in the edgeR package. The second method is the “Relative Log Expression” normalization (RLE) implemented in the DESeq2 package. The third method is the “Median Ratio Normalization” (MRN) (In Papyro Comparision of TMM (edgeR), RLE (DESeq2), and MRN Normalization Methods for a Simple Two-Conditions-Without-Replicates RNA-Seq Experimental Design). If the sample has biological replicates, we will preferentially use DESeq2 software.

  1. Multiple Testing Correction: The manuscript indicates the use of a P-value threshold of <0.05 for identifying differentially expressed genes (DEGs) and a P-value threshold of <0.01 for KEGG pathway enrichment analysis. However, it is crucial to address the issue of multiple testing to control for false positives. Applying appropriate multiple testing correction methods such as the Benjamini-Hochberg procedure for the DEG analysis and Bonferroni correction for the pathway enrichment analysis can help mitigate the risk of false discoveries.

Reply: Thanks for your thoughtful comments. For identification of differentially expressed genes (DEGs), we often used |log2FoldChange|>0, a P-value threshold of <0.05 or P-adj < 0.05. We also referred the related reference. Gong et al. identified differentially expressed genes using the statistical significance of the absolute value of |log2FoldChange| >0 and padj <0.01 (Transcriptomics-Based Study of Differentially Expressed Genes Related to Fat Deposition in Tibetan and Yorkshire Pigs). Zhang et al. used Padj < 0.05 and |log2FoldChange|>0 as the screening criteria for DEGs with biological duplication (Transcriptome Analysis Revealed a Positive Role of Ethephon on Chlorophyll Metabolism of Zoysia japonica under Cold Stress). Zhang et al. identified DEGs by using |log2foldchange|>0 and the corrected p-values < 0.05 as the screening criteria. We also re-identified DEGs by using |log2(fold change)|≥1, P-value<0.05 and P-adj <0.05 as the screening criteria. The related results have been added the attachment1. 

  1. Effect Size Estimation: In addition to using a log2(fold change) threshold of ≥1 to identify DEGs, it is essential to report effect size estimates and confidence intervals. Effect size measures provide a more interpretable representation of the magnitude of gene expression changes and allow readers to better understand the biological significance of the results.

Reply: Thanks for your thoughtful and valuable comments. For transcriptome sequencing, differentially expressed genes were defined by using P-value, q-value and log2(fold change). In addition, we also referred many references to identify DEGs using P-value, q-value and log2(fold change).

  1. Presentation of Results: The results of the differential expression analysis and pathway enrichment analysis should be presented in a clear and comprehensive manner. Providing volcano plots or heatmaps can be helpful in visualizing the distribution of DEGs and their functional enrichment.

Reply: Thanks for your valuable and thoughtful comments. The volcano plots were presented in Figure S1. The heatmap analysis of DEGs has added to Figure S2.

  1. Biological Interpretation: While GO and KEGG analyses are valuable tools for functional interpretation, it is crucial to provide a clear biological interpretation of the enriched pathways and their relevance to the insecticidal properties of coumarin. Discussion of the potential implications of the identified DEGs and pathways on the toxicological mechanisms would enhance the impact and significance of the study.

Reply: Thanks for your valuable comments. We have added the detailed descriptions in discussion.  

Reviewer 2 Report

Authors study the action of natural coumarin on Spodoptera litura. After the detrimental effects on the phenotype were observed, the transcriptome analysis was performed, the activity of tree detoxification enzymes was measured, effect of coumarin during CYP324A16 silencing was detected, and finally the metabolite profiling and affected metabolome pathways were reported. A lot of experimental data is collected and reported results are interesting.

Major claims regard the data presentation in the text.

In all figures some or majority of characters are too small and are not readable (exception are Figure 5 and figure 6, which are ok with the characters). I strongly recommend to increase the size of the figures and increase the resolution of the figures.

Abstract. Please, explain the enzyme and other abbreviations when first time mentioned (like AChE in line 15, and then GST, DEGs, TCA)

Introduction: In the list of affected insects, authors mention ref [11, Article in Chinese] where Plasmodium falciparum was affected. P.falciparum  is not insect, thus double check the relevance of this citation: Mu, L. Y.; Wang, Q. M.; Ni, Y. C. Effect of daphnetin on SOD activity and DNA synthesis of Plasmodium falciparum in vitro. Zhongguo Ji Sheng Chong Xue Yu Ji Sheng Chong Bing Za Zhi 2003, 21, 157-159

Results, Figure 1 B with Survival rate graph. In the figure legend for panel B “Macroscopic phenotype of larvae” is indicated, it looks like Panel B and panel C are confused in the legend.

Figure 1. When authors mention “Macroscopic phenotype”, please indicate in the Figure legend or in the Methods which phenotypic parameters exactly were observed.

Results, pg. 4. Line 121. Tree values for clean reads are reported for each condition. Please, explain in the text the meaning and the necessity of tree values (replications? ..something else?)

Results, pg. 4. Define all acronyms when first time used, as Q20, Q30 in line 128, GC in line 127, DESeq line 133, KEGG line 143.

Results, line 185. Gene P450B6 is mentioned. Which P450 cytochrome is it by CYP classification? Usually, one number is necessary before the letter B of cytochrome family.

Results. Table 2. Some DEGs in this Table and another Tables are identical (e.g. glutathione S-transferase 1 is present nine times in the Table 2 with different LOC number and cytochrome P450 6B6 – four times). Please, clearly declare in the text the presence in the tables the same/repeating enzymes. These enzymes are calculated as different entries, increasing the analysed DEG number, is it right? Very important that authors analyse the expression behaviour 0-24hrs-48hrs of the same enzymes for the coherence.

Results. Table 2. Conflicting data for glutathione S-transferase 1!

 LOC111349158 is changed UP from 695.0768 to 20166.12 while all another glutathione S-transferase 1 enzymes are changed DOWN (e.g. for LOC111355916 from 3748.005 to 1440.041). This result must be explained and discussed by authors!

Results, Figure 4 legend. Over the small non-readable characters in A4 printed version, the signs “a, b, c” for significance are confusing. It is not clear the significance between which data groups is reported. E.g. if I see “c” above first column in first upper-top diagram – what does it mean? The significance of differences between which groups is reported here?

Figure 6, panel A. In the scheme the meaning of four circles with small oval figures inside (RNAi? dsRNA?) is not clear. Please, indicate somewhere.

Methods. Describe precisely the “natural” coumarin provenience, supplier, purity etc.

Discussion. In larvae damaging experiments the concentrations 1%, 2% and 3% of coumarin was applied. Which coumarin concentration in normally detected in some typical food for Spodoptera?

Discussion. Authors have seen the strong impact on cytochrome CYP324A16. Please report all what is already known in the literature or databases about this CYP.

Conclusions. Line 594. Authors “proposed a hypothetical diagram illustrating that AChE and CYP450 activities were inhibited after coumarin…” Where is this diagram presented? How readers could see it?

Author Response

  1. In all figures some or majority of characters are too small and are not readable (exception are Figure 5 and figure 6, which are ok with the characters). I strongly recommend to increase the size of the figures and increase the resolution of the figures.

Reply: Thanks for your valuable comments. We have revised the related figures to increase the resolution in previous manuscript.

  1. Please, explain the enzyme and other abbreviations when first time mentioned (like AChE in line 15, and then GST, DEGs, TCA).

Reply: Thanks for your thoughtful suggestions. We have added the abbreviations in abstract.

  1. Introduction: In the list of affected insects, authors mention ref [11, Article in Chinese] where Plasmodium falciparum was affected. P.falciparum is not insect, thus double check the relevance of this citation: Mu, L. Y.; Wang, Q. M.; Ni, Y. C. Effect of daphnetin on SOD activity and DNA synthesis of Plasmodium falciparum in vitro. Zhongguo Ji Sheng Chong Xue Yu Ji Sheng Chong Bing Za Zhi 2003, 21, 157-159.

Reply: Thanks for your thoughtful comments. We have deleted the irrelevant reference in previous manuscript.

  1. Results, Figure 1 B with Survival rate graph. In the figure legend for panel B “Macroscopic phenotype of larvae” is indicated, it looks like Panel B and panel C are confused in the legend.

Reply: We are sorry for the incorrect descriptions in previous manuscript. We have revised the figure legend.

  1. Figure 1. When authors mention “Macroscopic phenotype”, please indicate in the Figure legend or in the Methods which phenotypic parameters exactly were observed.

Reply: Thanks for your thoughtful and valuable comments. We have added the related descriptions in Methods.

  1. Results, pg. 4. Line 121. Tree values for clean reads are reported for each condition. Please, explain in the text the meaning and the necessity of tree values (replications? ..something else?).

Reply: Thanks for your thoughtful and valuable comments. In the current study, three values for clean reads indicated that three values were obtained from three biological replications.

  1. Results, pg. 4. Define all acronyms when first time used, as Q20, Q30 in line 128, GC in line 127, DESeq line 133, KEGG line 143.

Reply: Thanks for your valuable comments. We have added the related definition for all acronyms.

  1. Results, line 185. Gene P450B6 is mentioned. Which P450 cytochrome is it by CYP classification? Usually, one number is necessary before the letter B of cytochrome family.

Reply: Thanks for your thoughtful comments. We have revised “P450B6” into “P4506B6” in previous manuscript.

  1. Table 2. Some DEGs in this Table and another Tables are identical (e.g. glutathione S-transferase 1 is present nine times in the Table 2 with different LOC number and cytochrome P450 6B6 – four times). Please, clearly declare in the text the presence in the tables the same/repeating enzymes. These enzymes are calculated as different entries, increasing the analysed DEG number, is it right? Very important that authors analyse the expression behaviour 0-24hrs-48hrs of the same enzymes for the coherence.

Reply: Thanks for your thoughtful comments. Based on transcriptome data, we found that different gene names exhibited the same annotation. Further analysis found that the mRNA sequences for identical glutathione S-transferase 1 or cytochrome P450 6B6 are different. We considered that these enzymes are calculated as different entries.  

  1. Table 2. Conflicting data for glutathione S-transferase 1!

Reply: Thanks for your thoughtful comments. In Table 2, for glutathione S-transferase 1, different gene names were annotated to same enzyme. We considered that these enzymes are calculated as different entries.

  1. LOC111349158 is changed UP from 695.0768 to 20166.12 while all another glutathione S-transferase 1 enzymes are changed DOWN (e.g. for LOC111355916 from 3748.005 to 1440.041). This result must be explained and discussed by authors!

Reply: Thanks for your thoughtful and valuable comments. Indeed, LOC111349158 encoding glutathione S-transferase 1 was significantly upregulated, while other genes downregulated. We speculated that different members belonging to GST1 have different functions. For example, Zou et al. revealed that glutathione S-transferase epilson 1 was upregulated in the midgut of S. litura when fed on diet containing indole-3-carbinol and allyl-isothiocyanate (Glutathione S-transferase SlGSTE1 in Spodoptera litura may be associated with feeding adaptation of host plants).

  1. Results, Figure 4 legend. Over the small non-readable characters in A4 printed version, the signs “a, b, c” for significance are confusing. It is not clear the significance between which data groups is reported. E.g. if I see “c” above first column in first upper-top diagram – what does it mean? The significance of differences between which groups is reported here?

Reply: Thanks for your thoughtful and valuable comments. We have revised the Figure 4 in previous manuscript.

  1. Figure 6, panel A. In the scheme the meaning of four circles with small oval figures inside (RNAi? dsRNA?) is not clear. Please, indicate somewhere.

Reply: We are sorry for unclear descriptions in previous manuscript. In panel A, the blue artificial diet, and the red oval symbol represented coumarin. We have added the related explanation in Figure 6 legend.

  1. Describe precisely the “natural” coumarin provenience, supplier, purity etc.

Reply: Thanks for your thoughtful comments. We have added the related information about the “natural” coumarin provenience, supplier and purity.

  1. In larvae damaging experiments the concentrations 1%, 2% and 3% of coumarin was applied. Which coumarin concentration in normally detected in some typical food for Spodoptera?

Reply: Thanks for your valuable comments. Spodoptera litura is an omnivorous herbivorous insect. In this study, we found that 1% coumarin had little effect on the growth and development of S. litura. Therefore, 3% coumarin was used to analyze the toxicological mechanism of coumarin on S. litura. In previous research, Wang et al. revealed that CarE activity increased 1.37 times in S. litura after 0.1% coumarin treatment. In fat body, P450 contend increased 14.5 times after 0.1% coumarin treatment (Effects of six plant secondary metabolites on activities of detoxification enzymes in Spodoptera litura).

  1. Authors have seen the strong impact on cytochrome CYP324A16. Please report all what is already known in the literature or databases about this CYP.

Reply: Thanks for your valuable comments. By consulting the relevant references, there are few reports on the functions of CYP324A16. Cheng et al. annotated 138 P450 genes in the S. litura genome, among which P450 clans 3 and 4 showed large expansions. CYP324A16 was attributed to clans 3, but its functions are unclear. Our results provide useful information for researching the functions of CYP324A16 in S. litura.

  1. Line 594. Authors “proposed a hypothetical diagram illustrating that AChE and CYP450 activities were inhibited after coumarin…” Where is this diagram presented? How readers could see it?

Reply: Thanks for your valuable and thoughtful comments. We have added a hypothetical diagram to previous manuscript.